# Representing Model Uncertainty of Neural Networks in Sparse Information Form

## Abstract

This paper presents a sparse representation of model uncertainty for deep neural networks (DNNs) that relies on an inverse formulation of Multivariate Normal Distribution (MND): an information form. We show that the model uncertainty can be estimated in this form using a scalable Laplace Approximation scheme, which involves a diagonal correction of the Kronecker-factored eigenbasis. As this makes the inversion of the information matrix intractable - an operation that is required for a full Bayesian analysis, we further devise a novel low-rank approximation of this eigenbasis that exploits spectral sparsity of DNNs. Methods to realize this sparsification are provided that develops into a memory-wise tractable sampling computations. Both of our theoretical analysis and empirical evaluations over various benchmarks show the superiority of our approach over existing methods.

## 1 Introduction

Whenever machine learning methods are used for safety-critical applications such as medical image analysis or autonomous driving, it is crucial to provide a precise estimation of the failure probability of the learned predictor. Therefore, most of the current learning approaches return distributions rather than single, most-likely predictions. For example, DNNs trained for classification usually use the softmax function to provide a distribution over predicted class labels. Unfortunately, this method tends to severely underestimate the true failure probability, leading to *overconfident* predictions (Guo et al., 2017). The main reason for this is that neural networks are typically trained with a principle of *maximum likelihood*, neglecting their *epistemic* or model uncertainty with the point estimates.

A widely known work by Gal (2016) shows that this can be mitigated by using dropout at test time. This so-called Monte-Carlo dropout (MC-dropout) has the advantage that it is relatively easy to use and therefore very popular in practice. However, MC-dropout also has significant drawbacks. First, it requires a specific stochastic regularization during training. This limits its use on already well trained architectures, because current networks are often trained with other regularization techniques such as batch normalization. Moreover, it uses a Bernoulli distribution to represent the complex model uncertainty, which in return, leads to an underestimation of the predictive uncertainty.

Several strong alternatives exist without these drawbacks. Variational inference (Khan et al., 2018; Kingma et al., 2015; Graves, 2011) and expectation propagation (Herandez-Lobato & Adams, 2015) are such examples. Yet, these methods use a diagonal covariance matrix which limits their applicability as the model parameters are often highly correlated. Building upon these, Sun et al. (2017); Louizos & Welling (2016); Zhang et al. (2018); Ritter et al. (2018a) show that the correlations between the parameters can also be computed efficiently by decomposing the covariance matrix of MND into Kronecker products of smaller matrices. However, not all matrices can be Kronecker decomposed and thus, these simplifications usually induce crude approximations (Bae et al., 2018). As the dimensionality of statistical manifolds are prohibitively too large in DNNs, more expressive, efficient but still easy to use ways of representing such high dimensional distributions are required.

To tackle this challenge, we propose to represent the model uncertainty in sparse information form of MND. As a first step, we devise a new Laplace Approximation (LA) for DNNs, in which we improve the state-of-the-art Kronecker factored approximations of the Hessian (George et al., 2018) by correcting the diagonal variance in parameter space. We show that these can be computed efficiently, and that the *information matrix* of the resulting parameter posterior is more accurate in terms of the Frobenius norm. In this way the model uncertainty is approximated in information form of the MND.

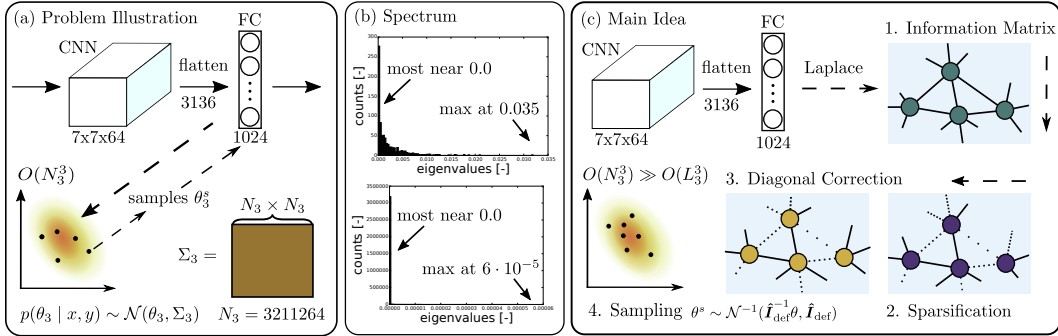

Figure 1: **Main idea.** (a) Covariance matrix $\Sigma$ for DNNs is intractable to infer, store and sample (an example taken from our MNIST experiments). (b) Our main insight is that the spectrum (eigenvalues) of information matrix (inverse of covariance) tend to be sparse. (c) Exploiting this insight a Laplace Approximation scheme is devised which applies a spectral sparsification (LRA) while keeping the diagonals exact. With this formulation, the complexity becomes tractable for sampling while producing more accurate estimates. Here, the diagonal elements (nodes in graphical interpretation) corresponds to information content in a parameter whereas the corrections (links) are the off-diagonals.

As this results in intractable inverse operation for sampling, we further propose a novel low-rank representation of the resulting Kronecker factorization, which paves the way to applications on large network structures trained on realistically sized data sets. To realize such sparsification, we propose a novel algorithm that enables a low-rank approximation of the Kronecker factored eigenvalue decomposition, and we demonstrate an associated sampling computations. Our experiments demonstrate that our approach is effective in providing more accurate uncertainty estimates and calibration on considered benchmark data sets. A detailed theoretical analysis is also provided for further insights.

We summarize our main contributions below.

- A novel Laplace Approximation scheme with a diagonal correction to the eigenvalue re-scaled approximations of the Hessian, as a practical inference tool (section 2.2).
- A novel low-rank representation of Kronecker factored eigendecomposition that preserves Kronecker structure (section 2.3). This results in a sparse information form of MND.
- A novel algorithm to enable a low rank approximation (LRA) for the given representation of MND (algorithm 1) and derivation of a memory-wise tractable sampler (section B.2).
- Both theoretical (section C) and experimental results (section 4) showing the applicability of our approach. In our experiments, we showcase the state-of-the-art performance within the class of Bayesian Neural Networks that are scalable and training-free.

To our knowledge we explore a sparse information form to represent the model uncertainty of DNNs for the first time. Figure 1 depicts our main idea which we provide more rigorous formulation next.

## 2 Methodology

### 2.1 Background and Notation

We model a neural network as a parameterized function $f_\theta : \mathbb{R}^{N_1} \to \mathbb{R}^{N_l}$ where $\theta \in \mathbb{R}^{N_\theta}$ are the weights and $N_\theta = N_1 + \cdots + N_l$. This function $f_\theta$ is in fact a concatenation of $l$ layers, where each layer $i \in \{1, ..., l\}$ computes $h_i = W_i a_{i-1}$ and $a_i = \phi(h_{i-1})$. Here, $\phi$ is a nonlinear function, $a_i$ are activations, $h_i$ linear pre-activations, and $W_i$ are weight matrices. The bias terms are absorbed into $W_i$ by appending 1 to each $a_i$. Thus, $\theta = \begin{bmatrix} vec(W_1)^T & vec(W_2)^T & ... & vec(W_l)^T \end{bmatrix}^T$ where $vec$ is the operator that stacks the columns of a matrix to a vector. Let $g_i = \delta h_i$, the gradient of $h_i$ w.r.t $\theta$.

Using LA the posterior is approximated with a Gaussian. The mean is then given by the MAP estimate $\theta_{MAP}$ and the covariance by the Hessian of the log-likelihood $(H + \tau I)^{-1}$ assuming a Gaussian prior with precision $\tau$. Using loss functions such as MSE or cross entropy and piece-wise linear activation $a_i$

(e.g RELU), a good approximation of the Hessian is the Fisher information matrix (IM) $I = \mathbb{E}\left[\delta\theta\delta\theta^T\right]$ for the backpropagated gradients $\delta\theta$ [1] and is typically scaled by the number of data points N (Martens & Grosse, 2015). IM is of size $N_\theta \times N_\theta$ resulting in too large matrix for moderately sized DNNs.

To make the computation tractable, a number of approximations are applied. First, it is assumed that the weights across layers are uncorrelated, which corresponds to a block-diagonal form of $I$ with blocks $I_1, I_2, \ldots, I_l$. Then, each realisation of block $I_i$ is represented as a Kronecker product $\delta\theta_i\delta\theta_i^T = a_{i-1}a_{i-1}^T \otimes g_i g_i^T$. This has an advantage that the inverse can be computed efficiently using $(A \otimes G)^{-1} = A^{-1} \otimes G^{-1}$. Then, matrices $A_{i-1}$ and $G_i$ are assumed to be statistically independent:

$$I_{i,kfac} = \mathbb{E}\left[a_{i-1}a_{i-1}^T \otimes g_i g_i^T\right] \approx \mathbb{E}\left[a_{i-1}a_{i-1}^T\right] \otimes \mathbb{E}\left[g_i g_i^T\right] = A_{i-1} \otimes G_i. \tag{1}$$

We refer to Martens & Grosse (2015) for details on KFAC. Here, $A_{i-1} \in \mathbb{R}^{n_i \times n_i}$ and $G_i \in \mathbb{R}^{m_i \times m_i}$, where the number of weights is $N_i = n_i m_i$. Applying equation 1 to LA (Ritter et al., 2018a), the parameter posterior per layer can be represented with a matrix normal distribution $\mathcal{MN}$ (defined in section A).

$$p(\theta_i \mid x, y) \sim \mathcal{N}(vec(W_{i,MAP}), I_{i,kfac}^{-1}) = \mathcal{MN}(W_{i,MAP}, A_{i-1}^{-1}, G_i^{-1}). \tag{2}$$

Typically IM is scaled by the number of data points N and incorporates the Gaussian prior $\tau$. The herein presented parameter posterior omits the addition of prior precision and scaling term for simplicity. Further note that, in practice, N and $\tau$ are treated as hyperparameters (Ritter et al., 2018a).

$$NI_i + \tau I \approx \left(\sqrt{N}A_{i-1} + \sqrt{\tau}I\right) \otimes \left(\sqrt{N}G_i + \sqrt{\tau}I\right). \tag{3}$$

KFAC scales to big data sets such as ImageNet (Krizhevsky et al., 2012) with large DNNs (Ba et al., 2017) and does not require changes in the training procedure when applied to LA.

## 2.2 Laplace Approximation with a diagonal correction

Following George et al. (2018), we first employ an eigenvalue correction in the Kronecker factored eigenbasis for LA. For simplicity, we drop layer indices $i$ and explanation herein applies layer-wise.

Let $I = V_{true}\Lambda_{true}V_{true}^T$ be the true eigendecomposition of IM per layer. From this it follows $\Lambda_{true} = \mathbb{E}\left[V_{true}^T\delta\theta\delta\theta^T V_{true}\right]$ and $\Lambda_{true,ii} = \mathbb{E}\left[(V_{true}^T\delta\theta)_i^2\right]$ for elements of layer wise matrices $i \in \{1, 2, \cdots, N\}$, Defining the eigendecomposition of $A$ and $G$ in equation 1 as $A = U_A S_A U_A^T$ and $G = U_G S_G U_G^T$, it further follows $I_{kfac} \approx A \otimes G = (U_A \otimes U_G)(S_A \otimes S_G)(U_A \otimes U_G)^T$ from the properties of the Kronecker product. Now, this approximation can be improved by replacing $(S_A \otimes S_G)$ with the eigenvalues $\Lambda_{true}$, where $V_{true}$ is set to $(U_A \otimes U_G)$ resulting in $\Lambda_{ii} = \mathbb{E}\left[[(U_A \otimes U_G)^T\delta\theta]_i^2\right]$. We denote it as EFB:

$$I_{efb} = (U_A \otimes U_G)\Lambda(U_A \otimes U_G)^T \text{ and } I_{efb}^{-1} = (U_A \otimes U_G)\Lambda^{-1}(U_A \otimes U_G)^T. \tag{4}$$

This technique has many desirable properties. Notably, it holds $\|I - I_{efb}\|_F \leq \|I - I_{kfac}\|_F$ wrt. the Frobenius norm as the computation is more accurate by correcting the diagonal in the eigenbasis.

However, there is an approximation in EFB since $(U_A \otimes U_G)$ is still an approximation of the true eigenbasis $V_{true}$. Intuitively, EFB only performs a correction of the diagonal elements in the eigenbasis, but when mapping back to the parameter space this correction is again harmed by the inexact estimate of the eigenvectors. Of course, an exact estimation of the eigenvectors is infeasible, but it is important to note that the diagonals of the exact IM $I_{ii} = \mathbb{E}\left[\delta\theta_i^2\right]$ can be computed efficiently using backpropagation. This and the fact that the off-diagonal elements of the IM are weaker with larger data sets or when using weight normalization (Neyshabur et al. (2016); Desjardins et al. (2015); Salimans & Kingma (2016)), motivates the idea to correct the approximation further as follows:

$$I_{def} = (U_A \otimes U_G)\Lambda(U_A \otimes U_G)^T + D \text{ where } D_{ii} = \mathbb{E}\left[\delta\theta_i^2\right] - \sum_{j=1}^{nm}(v_{i,j}\sqrt{\Lambda_j})^2. \tag{5}$$

---

[1]The expectation herein is defined wrt. the paramerterized density $p_\theta(y|x)$ assuming i.i.d. samples x.

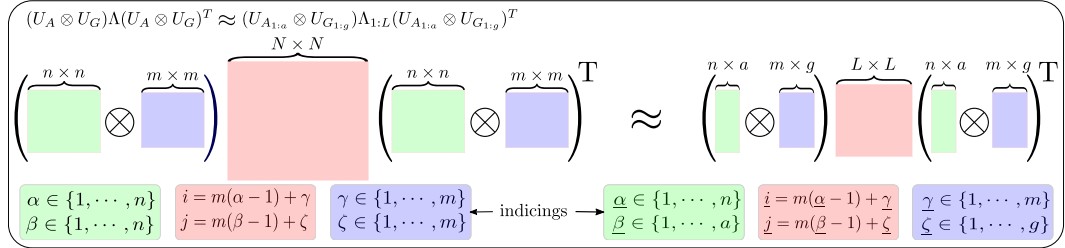

Figure 2: **Sparse Information Matrix.** We perform a low rank approximation on Kronecker factored eigendecomposition that preserves Kronecker structure in eigenvectors for two reasons: (a) reducing directly $(U_A \otimes U_G)_{1:L}$ is memory-wise infeasible, and (b) sampling scheme then only involves matrix multiplications of smaller matrices $U_{A_{1:a}}$ and $U_{G_{1:g}}$. Notations on indicing rules are also depicted.

In equation 5, we have represented $(U_A \otimes U_G)\Lambda(U_A \otimes U_G)_{ii}^T$ as $\sum_{k=1}^{nm}(v_{i,j}\sqrt{\Lambda_j})^2$ where $V = (U_A \otimes U_G) \in \mathbb{R}^{mn \times mn}$ is a Kronecker product with row elements $v_{i,j}$ (see definition 1 below). It follows from the properties of the Kronecker product that $i = m(\alpha - 1) + \gamma$. The derivation is shown in section B. Note that in this given form, the Kronecker products are never directly evaluated but the diagonal matrix $D$ can be computed recursively, making it computationally feasible.

**Definition 1:** *For $U_A \in \mathbb{R}^{n \times n}$ and $U_G \in \mathbb{R}^{m \times m}$, the Kronecker product of $V = U_A \otimes U_G \in \mathbb{R}^{mn \times mn}$ is given by $v_{i,j} = U_{a_{\alpha\beta}}U_{b_{\gamma,\zeta}}$, with the indices $i = m(\alpha - 1) + \gamma$ and $j = m(\beta - 1) + \zeta$. Here, the indices of the matrices $U_A$ and $U_G$ are $\alpha \in \{1, \cdots, n\}$, $\beta \in \{1, \cdots, n\}$, $\gamma \in \{1, \cdots, m\}$ and $\zeta \in \{1, \cdots, m\}$.*

Now, the parameter posterior distribution is represented in an information form or inverse formulation $\mathcal{N}^{-1}$ of MND as shown in equation 6 which is parameterized by an information vector $W_{MAP}^{IV} = I_{\text{def}}^{-1}vec(W_{MAP})$ and matrix $I_{\text{def}}$ from equation 5. Section A provides the formulation.

$$p(\theta \mid x, y) \sim \mathcal{N}(vec(W_{MAP}), I_{\text{def}}^{-1}) = \mathcal{N}^{-1}(W_{MAP}^{IV}, (U_A \otimes U_G)\Lambda(U_A \otimes U_G)^T + D) \qquad (6)$$

Unfortunately, in the current form, it involves a matrix inversion with size $N$ by $N$ when sampling. For some layers in modern architectures, this is not be feasible. This problem is tackled next.

## 2.3 REPRESENTING MODEL UNCERTAINTY IN SPARSE INFORMATION FORM

Sampling from the posterior is crucial. For example, an important use-case of the parameter posterior is estimating the predict uncertainty for test data $(x^*, y^*)$ by a full Bayesian analysis with $K_{mc}$ samples (equation 7). The herein approximation step is so-called Monte-carlo integration (Gal, 2016).

$$p(y^*|x^*, x, y) = \int p(y^*|x^*, \theta)p(\theta|x, y)d\theta \approx \frac{1}{K_{mc}}\sum_{t=1}^{K_{mc}} y^*(x^*, \theta_t^s) \text{ for } \theta^s \sim \mathcal{N}^{-1}(W_{MAP}^{IV}, I_{\text{def}}) \qquad (7)$$

However, directly sampling from equation 6 is non-trivial as explained in an example below.

**Example 1:** *Consider the architecture from figure 1 where the covariance matrix $\Sigma_3 \in \mathbb{R}^{N_3 \times N_3}$ for $N_3 = 3211264$. With equation 6, the sampling requires $O(N_3^3)$ complexity (the cost of inversion and finding a symmetrical factor) and obviously, this operation is computationally infeasible. Consequently, we next describe a sparse formulation of equation 6 that ensures tractability.*

To tackle this challenge, we propose the low rank form in equation 8[2] as a first step. Here, $\Lambda_{1:L} \in \mathbb{R}^{L \times L}$, $U_{A_{1:a}} \in \mathbb{R}^{m \times a}$ and $U_{G_{1:g}} \in \mathbb{R}^{n \times g}$ denote low rank form of corresponding eigenvalues and vectors (depicted in figure 2). Naturally, it follows that $L = ag$, $N = mn$ and furthermore, the persevered rank $L$ corresponds to preserving top $K$ and additional $J$ eigenvalues (resulting in $L \geq K$, $L = ag = K + J$).

$$I_{\text{def}} \approx \hat{I}_{\text{def}} = (U_{A_{1:a}} \otimes U_{G_{1:g}})\Lambda_{1:L}(U_{A_{1:a}} \otimes U_{G_{1:g}})^T + D \qquad (8)$$

---

[2]Note that the term D is added after LRA where D is computed similar to equation 5.

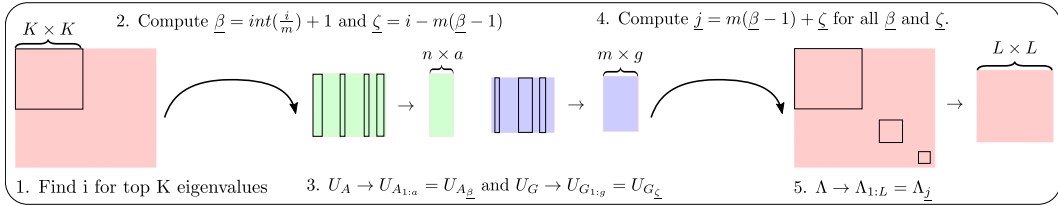

Figure 3: **Illustration of algorithm 1.** A low rank approximation on Kronecker factored eigende-composition that preserves Kronecker structure in eigenvectors constitutes steps 1 to 5.

Note the difference to preserving top $L$ eigenvalues and corresponding eigenvectors (Bishop, 2006) for LRA. In our case, this results in intractable $(U_A \otimes U_G)_{1:L}$ which defies the purpose. Therefore, as seen in equation 8, the Kronecker structure in eigenvectors as $(U_{A_{1:a}} \otimes U_{G_{1:g}})$ is preserved. Consequently, due to the Kronecker product operation, preserving top $K$ eigenvalues results in $L = K + J$ eigenvalues.

**Example 2:** *Let matrix $E$ decomposed as $E = U_{1:6}\Lambda_{1:6}U_{1:6}^T \in \mathbb{R}^{6\times6}$ with $U_{1:6} = \begin{bmatrix} u_1 & u_2 & \cdots & u_6 \end{bmatrix} \in \mathbb{R}^{6\times6}$ and $\Lambda_{1:6} = diag(\Lambda_1, \Lambda_2, \cdots, \Lambda_6) \in \mathbb{R}^{6\times6}$ in a descending order. In this toy example, the LRA with top 3 eigenvalues result in $E_{1:3} = U_{1:3}\Lambda_{1:3}U_{1:3}^T \in \mathbb{R}^{6\times6}$ (see notation to above). Instead, consider now the matrix $E_{kron} = (U_{A_{1:3}}\otimes U_{G_{1:2}})\Lambda_{1:6}(U_{A_{1:3}}\otimes U_{G_{1:2}})^T \in \mathbb{R}^{6\times6}$. Again, say we want to preserve top 3 the eigenvalues $\Lambda_{1:3}$ and corresponding eigenvectors $(U_{A_{1:3}}\otimes U_{G_{1:2}})_{1:3}$, However, as $(U_{A_{1:a}}\otimes U_{G_{1:g}})_{1:3} = \begin{bmatrix} u_{A_1} \otimes u_{G_1} & u_{A_1} \otimes u_{G_2} & u_{A_2} \otimes u_{G_1} \end{bmatrix}$, preserving the eigenvectors with the Kronecker structure results in having to store $U_{A_{1:2}} = \begin{bmatrix} u_{A_1} & u_{A_2} \end{bmatrix}$ and $U_{G_{1:2}} = \begin{bmatrix} u_{G_1} & u_{G_2} \end{bmatrix}$. Consequently, additional eigenvalue $\Lambda_4$ has to be saved in order to fulfill the definition of a Kronecker product $E_{kron_{1:3}} = (U_{A_{1:2}} \otimes U_{G_{1:2}})\Lambda_{1:4}(U_{A_{1:2}} \otimes U_{G_{1:2}})^T \in \mathbb{R}^{6\times6}$. In summary, preserving top $K$ eigenvalues results in other $J$ eigenvalues, which ensures the memory-wise tractability when performing LRA on large matrices.*

Then, how do we compute a low rank approximation that preserves Kronecker structures in eigenvectors? For this computation we propose algorithm 1 as an algorithmic contribution (also illustrated in figure 3). Let us start with a definition on indexing rules of Kronecker factored diagonal matrices.

**Definition 2:** *For diagonal matrices $S_A \in \mathbb{R}^{n\times n}$ and $S_G \in \mathbb{R}^{m\times m}$, the Kronecker product of $\Lambda = S_A \otimes S_G \in \mathbb{R}^{mn\times mn}$ is given by $\Lambda_i = s_{\alpha\beta}s_{\gamma\zeta}$, where the indices $i = m(\beta - 1) + \zeta$ with $\beta \in \{1, \cdots, m\}$ and $\zeta \in \{1, \cdots, n\}$. Then, given $i$ and $m$, $\beta = int(\frac{i}{m}) + 1$ and given $\beta$, $m$, and $i$, $\zeta = i - m(\beta - 1)$. Here, $int(\cdot)$ is an operator that maps its input to lower number integer.*

Notations in algorithm 1 are also depicted in figure 2. Now we explain with a toy example below.

**Example 3:** *For explaining algorithm 1, the toy example can be revisited. Firstly, as we preserve top 3 eigenvalues, $i \in \{1, 2, 3\}$ which are indices of eigenvalues $\Lambda_{1:3}$ (line 1). Then, using line 2, $\beta \in \{1, 2\}$ and $\zeta \in \{1, 2\}$ can be computed using definition 2. This relation holds as $\Lambda$ is computed from $S_A \otimes S_G$, and thus, $U_A$ and $U_G$ are their corresponding eigenvectors respectively. In line 3, we keep $U_{A_{1:2}}$ and $U_{G_{1:2}}$ using $\beta$ and $\zeta$. Again, in order to fulfill the Kronecker product operation, we use line 4 to find the eigenvalues $j \in \{1, 2, 3, 4\}$, and then preserve $\Lambda_{1:4}$. As explained, this has resulted in saving top 3 and additional 1 eigenvalues. Algorithm 1 provides the generalization of this and even if eigendecomposition does not come with a descending order, the same logic trivially applies.*

The incorporation of prior or regularization terms also follows without any additional approximation.

$$N\hat{I}_{\text{def}} + \tau I = (U_{A_{1:a}} \otimes U_{G_{1:g}})(N\Lambda_{1:L})(U_{A_{1:a}} \otimes U_{G_{1:g}})^T + (ND + \tau I) \tag{9}$$

**Sampling:** A key benefit of the proposed LRA is that now, sampling from the given covariance (equation 6 with the low rank form in equation 8; equation 9 with an incorporation of priors) only involves the inversion of a $L \times L$ matrix (in offline settings) and matrix multiplications of smaller Kronecker factored matrices or diagonal matrices during a full Bayesian analysis. To this end, we derive the analytical form of the sampler in section B.2 which makes the sampling computations feasible. This enables us to bound the intractable complexity of $O(N^3)$ to $O(L^3)$ for $L << N$.

**Algorithm 1:** Sparsification

**Input**: Matrices $U_A$, $U_G$, $\Lambda$ and Rank K.
**Output**: Matrices: $U_{A_{1:a}}$, $U_{G_{1:g}}$, $\Lambda_{1:L}$.
**Algorithm**:
1. Find indices of top K eigenvalues on $\Lambda$. This results in indices $i \in \{1, \cdots, K\}$.
2. For each elements of $i$, find corresponding indices of each Kronecker factors of original matrix $S_A \otimes S_G$ by using definition 2: $\underline{\beta} = int(\frac{i}{m}) + 1$ and $\underline{\zeta} = i - m(\underline{\beta} - 1)$. This results in indices of $\underline{\beta}$ and $\underline{\zeta}$ for $\bar{U}_A$ and $\bar{U}_G$ corresponding to $S_A \otimes S_G$ or $\Lambda_{1:K} = \bar{\Lambda}_i$.
3. Using obtained indices $\underline{\beta}$ and $\underline{\zeta}$, compute eigenvectors $U_{A_{1:a}} = \bar{U}_{A\underline{\beta}}$ and $\bar{U}_{G_{1:g}} = \bar{U}_{G\underline{\zeta}}$. These are the preserved eigenvectors in equation 8.
4. Find indices of top K and additional J eigenvalues using $\underline{j} = m(\underline{\beta} - 1) + \underline{\zeta}$ for all $\underline{\beta}$ and $\underline{\zeta}$.
5. Preserve eigenvalues $\Lambda_{1:L} = \bar{\Lambda}_{\underline{j}}$ where $\underline{j} \subseteq j$. $\Lambda_{1:L}$ represents the preserved top K and additional J eigenvalues in equation 8.

**Algorithm 2:** Inference of IM

**Input**: Pre-trained Neural Network and train data.
**Output**: Matrices: $U_{A_{1:N}}$, $U_{G_{1:M}}$, $\Lambda_{1:L}$ and $D$.
**Algorithm**:
**for** *the given data points* **do** *KFAC*
    **for** $i := 1$ *to* $l$ **do**
        | Compute $A_i$ and $G_i$ (equation 1).
    **end**
**end**
Compute $U_A$, $U_G$ with eigenvalue decomposition.
**for** *the given data points* **do** *EFB*
    **for** $i := 1$ *to* $l$ **do**
        Compute $\mathbb{E}\left[\delta\theta_i^2\right]$.
        Compute $\Lambda_{ii}$ with $U_A$ and $U_G$.
    **end**
**end**
**for** *all the layers* **do** *DEF (without involving data)*
    Compute $U_{A_{1:N}}$, $U_{G_{1:M}}$, $\Lambda_{1:L}$ (algorithm 1).
    Compute $D$ (equation 5).
**end**

**Overview:** An overview is depicted in figure 1 where we first show that IM of DNNs tend to be sparse in its spectrum (similar to the findings of Sagun et al. (2018)). With this insight we propose to represent the parameter posterior in a sparse information form which is visualized with its graphical interpretations. From IM of EFB, we apply our LRA that weakens the strengths of weak nodes (diagonals of IM) and links (off-diagonals) in a preserving fashion. Then, a diagonal correction can be added to keep the information of each nodes exact. A key benefit is that the sampling computations can be achieved in a memory-wise feasible way. Algorithm 2 shows the overall procedures. Further note that, as IM is estimated after training, our method can be applied to existing architectures. EFB is also computed in a different way to George et al. (2018) so that our EFB does not require batch assumption for taking expectations, and the scheme is cheaper since eigenvalue decomposition of $A_{i-1}$ and $G_i$ are computed only once. Computing diagonal correction term also does not involve data.

$$p(\theta \mid x, y) \sim \mathcal{N}^{-1}(\hat{W}_{MAP}^{IV}, (U_{A_{1:a}} \otimes U_{G_{1:g}})\Lambda_{1:L}(U_{A_{1:a}} \otimes U_{G_{1:g}})^T + D) \tag{10}$$

As a result our approach yields a sparse information form of MND where the IM has a low rank eigendecomposition plus diagonal structure that preserves Kronecker structure in eigenvectors (shown above; prior and scaling terms are omitted to keep the notation uncluttered; $\hat{W}_{MAP}^{IV}$ is an information vector associated to the proposed IM). Since this formulation of model uncertainty has not bee studied before, we provide theoretical results in section C for further insights and justifications.

## 3 RELATED WORKS

**Sparse Information Filters:** A similar idea of sparsifying the information matrix while keeping the diagonals accurate can be found in sparse information filters. Here, Bayesian tracking is realized in information form of MND instead of canonical counterparts (Kalman Filters). As this leads to inefficiency in marginalization, sparsity is introduced while keeping the diagonals accurate (Thrun et al., 2004). A main difference, however, is that DNNs typically have higher dimensions and a sparse structure in the spectrum (eigenvalues) in contrast to spaces of parameters in SLAM problems. Thus, we propose to explore Kronecker factorization and induce spectral sparsity or LRA respectively.

**Approximation of the Hessian:** The Hessian of DNNs is prohibitively too large as its size is quadratic to the parameter space. For this problem an efficient approximation is a layer-wise Kronecker factorization (Martens & Grosse, 2015; Botev et al., 2017) which have demonstrated a notable scalability (Ba et al., 2017). In a recent extension of (George et al., 2018) the eigenvalues of the Kronecker factored matrices are re-scaled so that the diagonal variance in its eigenbasis is exact. The work demonstrates a provable method of achieving higher accuracy. Yet, as this is harmed by inaccurate estimates of eigenvectors, we further correct the diagonals in the parameter space.

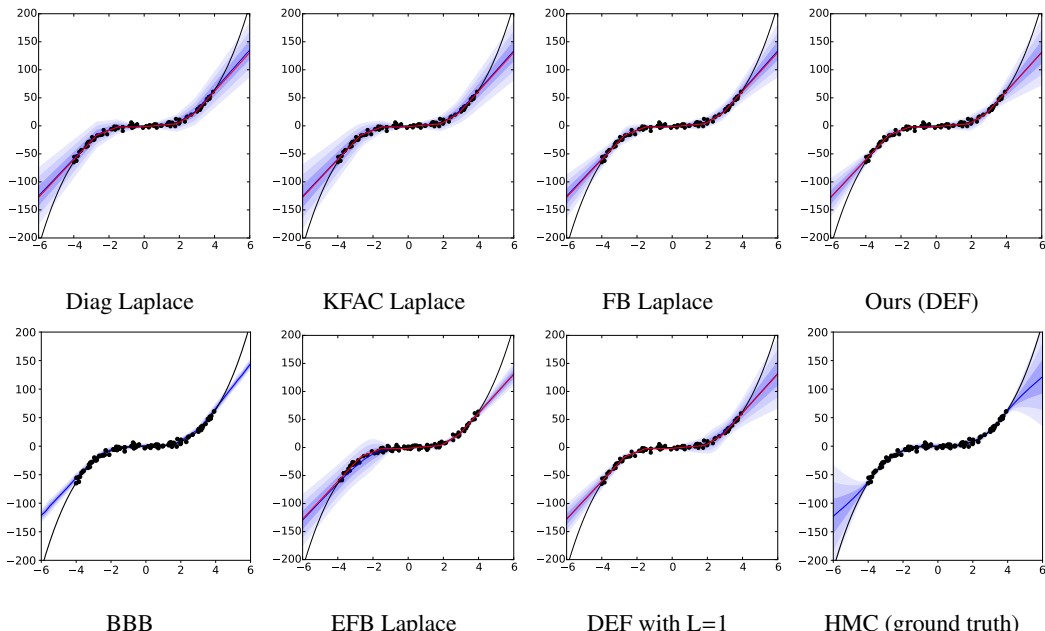

Figure 4: **Uncertainty on toy regression.** The black dots and the black lines are data points (x, y). The red and blue lines show predictions of the deterministic Neural Network and the mean output respectively. Upto three standard deviations are shown with blue shades.

**Laplace Approximation:** Instead of methods rooted in variational inference (Hinton & van Camp, 1993) and sampling (Neal, 1996), we build upon LA (MacKay, 1992) as a practical inference framework. Recently, diagonal (Becker & Lecun, 1989) and Kronecker-factored approximations (Botev et al., 2017) to the Hessian have been applied to LA by Ritter et al. (2018a). The authors have further proposed to use LA in continual learning (Ritter et al., 2018b), and demonstrate a competitive results by significantly outperforming its benchmarks (Kirkpatrick et al., 2017; Zenke et al., 2017). Building upon Ritter et al. (2018a) for approximate inference, we propose to use more expressive posterior distribution than matrix normal distribution. In the context of variational inference, SLANG (Mishkin et al., 2018) share similar spirit to ours in using a low-rank plus diagonal form of covariance where the authors show the benefits of low-rank approximation in detail. Yet, SLANG is different to ours as they do not explore Kronecker structures and requires changes in the training procedure.

**Dimensionality Reduction:** A vast literature is available for dimensionality reduction beyond principal component analysis (Wold et al., 1987) and singular value decomposition (Golub & Reinsch, 1971; Van Der Maaten et al., 2009). To our knowledge though, dimensionality reduction in Kronecker factored eigendecomposition that maintains Kronecker structure of eigenvectors has not been studied before. Thus, we propose algorithm 1 and further provide its theoretical properties in section C.

## 4 EXPERIMENTAL RESULTS

An empirical study is presented with a toy regression and classification tasks across MNIST (Lecun et al., 1998), notMNIST (Bulatov, 2011), CIFAR10 (Krizhevsky, 2009) and SHVN (Netzer et al., 2011) data-sets. The experiments are designed to demonstrate the quality of predictive uncertainty, effects of varying LRA, the quality of approximate Hessian, and gains in reduction of computational complexity due to LRA. All experiments are implemented using Tensorflow (Abadi et al., 2016).

### 4.1 TOY REGRESSION AND EFFECTS OF LOW RANK APPROXIMATION

**Predictive Uncertainty:** Firstly, an evaluation on toy regression data-set is presented. This experiment has an advantage that we can not only evaluate the quality of predictive uncertainty, but also directly compare various approximations to the Hessian. For this a single-layered fully connected

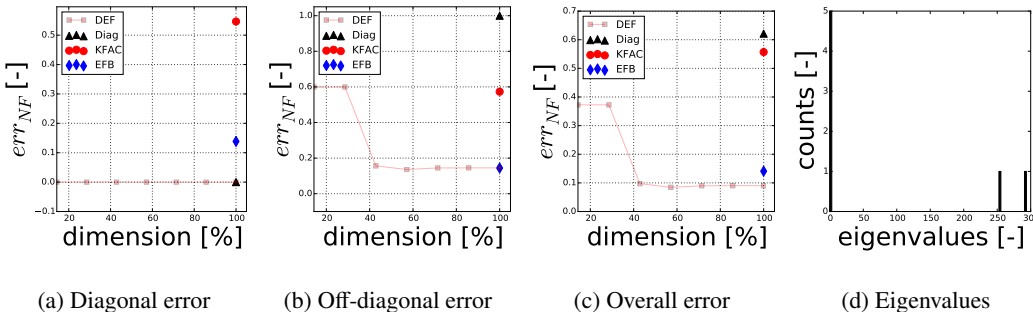

| (a) Diagonal error | (b) Off-diagonal error | (c) Overall error | (d) Eigenvalues |

Figure 5: **Effects of Low Rank Approximation** in Frobenius norm of error. This measure is normalized from 0 to 1. Lower the better. Laplace based methods such as EFB, Diag, KFAC and DEF are compared in terms of diagonal, off-diagonal and overall resulting approximation error to exact block diagonal hessian. Eigenvalue histogram is also plotted.

network with seven units in the first layer is considered. We have used 100 uniformly distributed points $x \sim U(-4, 4)$ and samples $y \sim N(x^3, 3^2)$. Visualization of predictive uncertainty is shown in figure 4. HMC (Neal, 1996), BBB (Blundell et al., 2015), diagonal and KFAC Laplace Ritter et al. (2018a) have been included as a comparison whereas EFB Laplace, exact block diagonal Hessian (FB) and DEF with a full rank and one rank are presented for an ablation study. Both Diag and KFAC Laplace are tuned similar to Ritter et al. (2018a) by regularizing them. DEF variants and FB Laplace for this experiment did not require hyper-parameter tuning. Implementation details are in section E.

All the methods show higher uncertainty in the regimes far away from training data where BBB showing the most difference to HMC. Furthermore, Diag, KFAC and EFB Laplace predicts rather high uncertainty even within the regions that are covered by the training data. DEF variants slightly underestimate the uncertainty but produces the most comparable fit to the FB Laplace and HMC (our ground truths). We believe this is the direct effect of modelling the Hessian more accurately [3]. Moreover, since the only difference between EFB and DEF Laplace is a diagonal correction term, this empirical results suggest that keeping diagonals of IM exact results in accurate predictive uncertainty.

**Effects of Low Rank Approximation:** Next, we quantitatively study the effects of LRA by directly evaluating on the approximations of IM. This is because uncertainty estimation, despite being a crucial entity, are confounded from the problem itself and may not reveal the algorithmic insights to its full potential. For this, we revisit the toy regression problem and provide a direct evaluation of IM with measure on normalized Frobenius norm of error $err_{NF}$ in the first layer of the network.

The results are shown in figure 5. Here, the reduced dimension is not proportional to the ranks (e.g. many zero or close to eigenvalues). Figure 5 (a) depicts that DEF results in accurate estimates on $\boldsymbol{I}_{ii}$ regardless of the chosen dimensions $L$ while EFB has the more approximation error, which we believe is due to inaccurate estimates of eigenvectors. KFAC on the other hand, produces the most errors on diagonal elements, which indicate that its assumption of Kronecker factorization induces crude approximation in this experiment. Regarding the off-diagonal errors EFB also outperforms KFAC and Diag estimates. Furthermore, error profile of off-diagonal error $\boldsymbol{I}_{ij}$ also explains the principles of the LRA that as we decrease the ranks, the error increases but in a preserving manner. These results can also be explained by *Lemma 1 and 4* of section C which reflects the design principles of the method.

### 4.2 Classification and Reduction in Complexity

**Predictive Uncertainty:** Next, we evaluate predictive uncertainty on classification tasks in which the proposed low-rank representation is strictly necessary. Furthermore, our goal is not to achieve the highest accuracy but evaluate predictive uncertainty. To this end, we choose classification tasks with known and unknown classes, e.g. a network is not only trained and evaluated on MNIST but also tested using notMNIST. Note that under such tests, any probabilistic methods should report their evaluations on both known and unknown classes with the same hyperparameter settings. This is because a Bayesian Neural Network to be always highly uncertain, which may seem to work well on

---

[3] we comment on this statement, and the effects of data-set size to number of parameters in section E.

out-of-distribution samples but are always overestimating uncertainty, even for the correctly classified samples within the distribution similar to the train data. For evaluating predictive uncertainty on known classes, Expectation Calibration Error (ECE) has been used. As we found it more intuitive, normalized entropy is reported for evaluating predictive uncertainty on unknown classes.

Table 1: **Results of classification experiments.** Accuracy and ECE are evaluated on in-domain distribution (MNIST and CIFAR10) whereas entropy is evaluated on out-of-distribution (notMNIST and SHVN). Lower the better for ECE. Higher the better for entropy and accuracy.

| MNIST | NN | Diag | KFAC | MC-dropout | Ensemble | EFB | DEF |
|---|---|---|---|---|---|---|---|
| *Accuracy* | 0.993 | 0.9935 | 0.9929 | 0.9929 | **0.9937** | 0.9929 | 0.9927 |
| *ECE* | 0.395 | 0.0075 | 0.0078 | 0.0105 | 0.0635 | 0.012 | **0.0069** |
| *Entropy* | 0.055±0.133 | 0.555 ± 0.196 | 0.599 ± 0.199 | 0.562 ± 0.19 | 0.596 ± 0.133 | 0.618 ± 0.185 | **0.635 ± 0.19** |
| CIFAR | NN | Diag | KFAC | MC-dropout | Ensemble | EFB | DEF |
| *Accuracy* | 0.8606 | **0.8659** | 0.8572 | N/A | 0.8651 | 0.8638 | 0.8646 |
| *ECE* | 0.0819 | 0.0358 | 0.0351 | N/A | 0.0809 | 0.0343 | **0.0084** |
| *Entropy* | 0.245 ± 0.215 | 0.4129 ± 0.197 | 0.408 ± 0.197 | N/A | 0.370 ± 0.192 | 0.417 ± 0.196 | **0.4338 ± 0.18** |

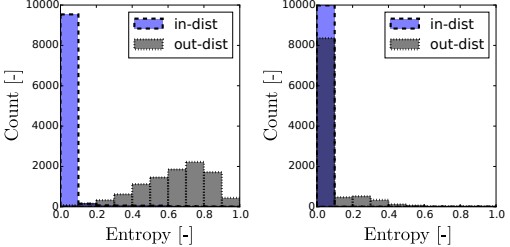

| MNIST | Dim N [-] | Dim L [-] | Percent [%] |
|---|---|---|---|
| *CNN-1* | 800 | 450 | **56.25** |
| *CNN-2* | 51200 | 5185 | **10.12** |
| *FC-1* | 3211264 | 5625 | **0.18** |
| *FC-2* | 10240 | 4775 | **46.63** |
| CIFAR | Dim N [-] | Dim L [-] | Percent [%] |
| *CNN-1* | 4800 | 4800 | **100** |
| *CNN-2* | 102400 | 2112 | **2.06** |
| *FC-1* | 884736 | 3980 | **0.45** |
| *FC-2* | 73728 | 5499 | **7.45** |
| *FC-3* | 1920 | 1920 | **100** |

Figure 6: **Normalized Entropy histogram** (left: DEF Laplace and right: deterministic) on MNIST vs notMNIST experiments. Our method clearly separates the out-of-distribution and wrongly classified samples (out-dist) to the correctly classified samples from in-domain distribution (in-dist).

Table 2: **Reduction in complexity.** Reduced dimensions from N to the chosen rank L per layer are reported for both MNIST and CIFAR experiments. CNN stand for convolution while FC is for fully connected layers. The complexity of sampling computations $O(N^3)$ are reduced to $O(L^3)$.

On MNIST-notMNIST experiments, we compare to MC-dropout (Gal, 2016), ensemble (Lakshminarayanan et al., 2017) of size 15, Diag and KFAC Laplace (Ritter et al., 2018a). These methods are state-of-the-art baselines that have a merit of requiring no changes in the training procedure. The later is crucial for a fair comparison as we can use the same experiment settings (Mukhoti et al., 2018). Regarding the architectures, LeNet with RELU and a L2 coefficient of 1e-8 has been the choice. In particular, this typically makes a neural network overconfident, and we can see the effects of model uncertainty. This architecture validates our claim as it has the parameters of size $\theta_3 \in \mathbb{R}^{3137 \times 1024}$ in the $3^{rd}$ layer. Obviously, its covariance is intractable as it is quadratic in size (see figure 1). The results can be found in table 1. Firstly all the methods improved significantly over the deterministic one (denoted NN). Furthermore, DEF Laplace achieved here the lowest ECE, at the same time, predicted with the highest mean entropy on out-of-distribution samples. Figure 6 shows this result where our method separates between wrong and correct predictions which stems from the domain change.

Further tests were performed on CIFAR10 (known) and SVHN (unknown) to see the generalization under batch normalization and data augmentation. For this, we trained a 5 layer architecture with 2 CNN and 3 FC layers. The results are also reported in table 1. Similar to MNIST experiments, our method resulted in a better calibration performance and out-of-distribution detection overall. Note that for Diag, KFAC and EFB Laplace, grid searches on hyperparameters were rather non-trivial here. Increasing $\tau I$ had the tendency to reduce ECE on CIFAR10, but in return resulted in underestimating the uncertainty on SVHN and vice versa. DEF Laplace instead, required smallest regularization hyperparameters to strike a good balance between these two objectives. We omitted dropout as using it as a stochastic regularization instead of batch normalization would result in a different network and thus, comparison would be not meaningful. More implementation details are provided in section E.

**Reduction in Complexity:** The proposed LRA has been imposed as a means to tackle the challenges of computational intractability of MND. To empirically access the reduction in complexity, we depict the parameter and low rank dimensions N and L respectively in table 2. As demonstrated, our LRA

based sampling computations reduce the computational complexity significantly. Furthermore, this explains the necessity of LRA - certain layers (e.g. FC-1 of both MNIST and CIFAR experiments) are computationally intractable to store, infer and sample. As a result, we demonstrate an alternative representation for DNNS without resorting to fully factorized and matrix normal distribution.

**Discussion and Limitations:** Importantly, we demonstrate that when projected to different success criteria, no inference methods largely win uniformly. Yet these experiments also show empirical evidence that our method works in principle and compares well to the state-of-the-art. Representing layer-wise MND in a sparse information form, and demonstrating a low rank inverse/sampling computations, we show an alternative approach of designing scalable and practical inference framework. Finally, these results also indicate that keeping the diagonals of IM accurate while sparsifying the off-diagonals can lead to outstanding performance in terms of predictive uncertainty and generalizes well to various data, models and even measures. For future works, we share the view that comparing different approximations to the true posterior is quite challenging for DNNs. Consequently, better metrics and benchmarks that show the benefits of model uncertainty can be an important direction.

On the other hand, we also address a key limitation of our work which stems from two hypothesis: (a) when represented in information form, the spectrum of IM should be sparse, and (b) keeping the diagonals exact while sparsifying the off-diagonals should result in a better estimates of model uncertainty (equivalently keeping the information content of a node exact while sparsifying the weak links between the nodes from a graphical interpretation of information matrix). While empirical evidence from prior works (Sagun et al., 2018; Thrun et al., 2004; Bailey & Durrant-Whyte, 2006) along with our experiments validate these to some extent, there exists no theoretic guarantees to our knowledge. Consequently, theoretical studies that connect information geometry (Amari, 2016) of DNNs and Bayesian Neural Networks can be an exciting venue of future research. Nevertheless, similar to sparse Gaussian Processes (Snelson & Ghahramani, 2006), we believe our work can be a stepping stone for sparse Bayesian Neural Networks that goes beyond approximate inference alone.

## 5 CONCLUSION

We address an effective approach of representing model uncertainty in deep neural networks using Multivariate Normal Distribution, which has been thought computationally intractable so far. This is achieved by designing its novel sparse information form. With one of the most expressive representation of model uncertainty in current Bayesian deep learning literature, we show that uncertainty can be estimated more accurately than existing methods. For future works, we plan to demonstrate a real world application of this approach, pushing beyond the validity of concepts.

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

# A  Notations on Distributions

## A.1  Matrix Normal Distribution

The matrix normal distribution is a probability density function for the random variable $X \in \mathbb{R}^{n \times m}$ in matrix form. It can be parameterized with mean $W_{\text{MAP}} \in \mathbb{R}^{n \times m}$, scales $U_A \in \mathbb{R}^{n \times n}$ and $U_G \in \mathbb{R}^{m \times m}$. It is essentially a multivariate Gaussian distribution with mean $\text{vec}(W_{\text{MAP}})$ and covariance $U \otimes V$.

$$p(X|W_{\text{MAP}}, U_A, U_G) = \frac{\exp\left(-\frac{1}{2}\text{tr}\left[U_G^{-1}(X - W_{\text{MAP}})^T U_A^{-1}(X - W_{\text{MAP}})\right]\right)}{(2\pi)^{\frac{nm}{2}}|U_G|^{\frac{n}{2}}|U_A|^{\frac{m}{2}}} \tag{11}$$

In section B, we denote this distribution with $\mathcal{MN}$ parameterized by $W_{\text{MAP}}$, $U_A$ and $U_G$ so that $p(X|W_{\text{MAP}}, U_A, U_G) = \mathcal{MN}(W_{\text{MAP}}, U_A, U_G)$. Here, tr stands for trace operation and we omitted layer indicing i for better clarity. Refer to Gupta & Nagar (1999) for more details.

## A.2  Information Form of Multivariate Normal Distribution

Information form of Multivariate Normal Distribution (MND) is a dual representation for the well known canonical form. Lets denote $\bar{x} = \text{vec}(X) \in \mathbb{R}^{nm}$, $\mu = \text{vec}(W_{\text{MAP}}) \in \mathbb{R}^{nm}$ and $\Sigma = \boldsymbol{I}^{-1} \in \mathbb{R}^{mn \times nm}$ as a random variable, mean and covariance respectively for $N = mn$.

$$p(\bar{x}|\mu, \Sigma) \propto \exp\left\{-\frac{1}{2}\bar{x}^T \Sigma^{-1} \bar{x} + \mu^T \Sigma^{-1} \mu\right\} \tag{12}$$

Then, equation 12 defines the canonical form. Now we denote its Information form in equation 13.

$$p(\bar{x}|b, F) \propto \exp\left\{-\frac{1}{2}\bar{x}^T \boldsymbol{I} \bar{x} + W_{\text{MAP}}^{IV} \bar{x}\right\} \tag{13}$$

Here, $\bar{x} \in \mathbb{R}^{mn}$ represent the random variable as well. $W_{\text{MAP}}^{IV} = \Sigma^{-1}\mu \in \mathbb{R}^{mn}$ and $\boldsymbol{I} = \Sigma^{-1} \in \mathbb{R}^{mn \times mn}$ are information vector (denoted IV in the main text with superscript) and matrix respectively. We denote the information form as $\mathcal{N}^{-1}$ which is completely described by an information vector and matrix. Information matrix is also widely known as precision matrix. Thrun et al. (2004) in Simultaneous Localization and Mapping (SLAM) literature provides a good overview and explanations.

# B  Derivations

## B.1  Derivation 1: Diagonal correction without evaluating the Kronecker products

Directly evaluating $U_A \otimes U_G$ may not be computationally feasible for modern DNNs. Therefore, we derive the analytical form of the diagonal elements for $(U_A \otimes U_G)\Lambda(U_A \otimes U_G)^T$ without having to fully evaluate it. Let $U_A \in \mathbb{R}^{n \times n}$ and $U_G \in \mathbb{R}^{m \times m}$ be the square matrices. $\Lambda \in \mathbb{R}^{mn \times mn}$ is a diagonal matrix by construction. $V = U_A \otimes U_G \in \mathbb{R}^{mn \times mn}$ is a Kronecker product with elements $v_{i,j}$ with $i = m(\alpha - 1) + \gamma$ and $j = m(\beta - 1) + \zeta$ (from definition of Kronecker product). Then, the diagonal entries of $(U_A \otimes U_G)\Lambda(U_A \otimes U_G)^T$ can be computed as follows:

$$\left[(U_A \otimes U_G)\Lambda(U_A \otimes U_G)^T\right]_{ii} = \sum_{j=1}^{nm}(v_{i,j}\sqrt{\Lambda_j})^2 \tag{14}$$

**Derivation:** As a first step of the derivation, we express $(A \otimes B)\Lambda(A \otimes B)^T$ in the following form:

$$\begin{aligned}
(U_A \otimes U_G)\Lambda(U_A \otimes U_G)^T &= (U_A \otimes U_G)\Lambda^{\frac{1}{2}}\Lambda^{\frac{1}{2}}(U_A \otimes U_G)^T \\
&= \left[(U_A \otimes U_G)\Lambda^{\frac{1}{2}}\right]\left[(U_A \otimes U_G)\Lambda^{\frac{1}{2}}\right]^T \\
&= UU^T
\end{aligned} \tag{15}$$

Then, $\text{diag}(UU^T)_i = \left[UU^T\right]_{ii} = \sum_{j=1}^{nm} u_{ij}^2$ by definition. Now, we let $(U_A \otimes U_G)\Lambda^{\frac{1}{2}} = V\Lambda^{\frac{1}{2}}$ with $\Lambda^{\frac{1}{2}}$ being again a diagonal matrix. Therefore, $u_{ij} = v_{i,j}\sqrt{\Lambda_j}$ due to the multiplication with a diagonal matrix from a right hand side. Substituting back these results in $\left[(U_A \otimes U_G)\Lambda(U_A \otimes U_G)^T\right]_{ii} = \sum_{j=1}^{nm}(v_{i,j}\sqrt{\Lambda_j})^2$ which completes the derivation. Formulating equation 14 for the non-square matrices (which results after a low rank approximation) such as $U_{A_{1:a}} \in \mathbb{R}^{n \times a}$ and $U_{G_{1:g}} \in \mathbb{R}^{m \times g}$ and paralleling this operation are rather trivial and hence, they are omitted.

## B.2 Derivation 2: Analytical Derivation of a Sampler

For a full Bayesian analysis which is approximated by a Monte Carlo integration, sampling is a crucial operation (see equation 7) for computing predictive uncertainty. We start by stating the problem.

**Problem statement:** Consider drawing samples $\text{vec}(W^s) \in \mathbb{R}^{nm}$ from our sparse information form:

$$vec(W^s) \sim \mathcal{N}^{-1}(W_{\text{MAP}}^{IV}, (U_{A_{1:a}} \otimes U_{G_{1:g}})\Lambda_{1:L}(U_{A_{1:a}} \otimes U_{G_{1:g}})^T + D) \tag{16}$$

Typically, drawing such samples $\text{vec}(W^s)$ from a canonical form of MND requires finding a symmetrical factor of the covariance matrix (e.g. Chloesky decomposition) which is cubic in cost $O(N^3)$. Even worse, when represented in an information form as in equation 16, it requires first an inversion of information matrix and then the computation of a symmetrical factor which overall constitutes two operations of cost $O(N^3)$. Clearly, if $N$ lies in a high dimension such as 1 million, even storing is obvious not feasible, let alone the sampling computations. Therefore, we need a sampling computation that (a) keeps the Kronecker structure while sampling so that first, the storage is memory-wise feasible, and then (b) the operations that require cubic cost such as inversion, must be performed in the dimensions of low rank L instead of full parameter dimensions N. We provide the solution below.

**Analytical solution:** Let us define $X^l \in \mathbb{R}^{mn}$ and $X^s \in \mathbb{R}^{m \times n}$ as the samples from a standard Multivariate Normal Distribution in equation 17 where we denote the followings: $0_{nm} \in \mathbb{R}^{nm}$, $I_{mn} \in \mathbb{R}^{mn \times mn}$, $0_{n \times m} \in \mathbb{R}^{n \times m}$, $I_n \in \mathbb{R}^{n \times n}$ and $I_m \in \mathbb{R}^{m \times m}$. Note that these sampling operations are cheap.

$$X^l \sim N(0_{nm}, I_{nm}) \text{ or } X^s \sim \mathcal{MN}(0_{n \times m}, I_n, I_m). \tag{17}$$

Furthermore, we denote $W^l = vec(W^s) \in \mathbb{R}^{mn}$, $\theta_{\text{MAP}} = vec(W_{\text{MAP}}) \in \mathbb{R}^{mn}$ as a sample from equation 16 and its mean as a vector respectively. We also note that $\Lambda_{1:L} \in \mathbb{R}^{L \times L}$ and $D \in \mathbb{R}^{mn \times mn}$ are the low ranked form of the re-scaled eigen-values and the diagonal correction term as previously defined. $U_{A_{1:a}} \in \mathbb{R}^{m \times a}$ and $U_{G_{1:g}} \in \mathbb{R}^{n \times g}$ are the eigenvectors of low ranked eigen-basis so that $m \geq a$, $n \geq g$ and $L = ag$. Then, the samples of 16 can be computed analytically as[4]:

$$W^l = \theta_{\text{MAP}} + F^c X^l \text{ where,}$$
$$F^c = D^{-\frac{1}{2}}\Big(I_{nm} - D^{-\frac{1}{2}}(U_{A_{1:a}} \otimes U_{G_{1:g}})\Lambda_{1:L}^{\frac{1}{2}}(C^{-1} + V_s^T V_s)^{-1}\Lambda_{1:L}^{\frac{1}{2}}(U_{A_{1:a}} \otimes U_{G_{1:g}})^T D^{-\frac{1}{2}}\Big). \tag{18}$$

Firstly, the symmetrical factor $F^c \in \mathbb{R}^{mn \times mn}$ in equation 18 is a function of matrices that are feasible to store as they involve diagonal matrices or small matrices in a Kronecker structure. Furthermore,

$$V_s = D^{-\frac{1}{2}}(U_{A_{1:a}} \otimes U_{G_{1:g}})\Lambda_{1:L}^{\frac{1}{2}}$$
$$C = A_c^{-T}(B_c - I_L)A_c^{-1} \text{ with } A_c \text{ and } B_c \tag{19}$$

being the Cholesky decomposed matrices of $V_s^T V_s \in \mathbb{R}^{L \times L}$ and $V^T V + I_L \in \mathbb{R}^{L \times L}$ such that:

$$A_c A_c^T = V_s^T V_s \text{ and}$$
$$B_c B_c^T = V_s^T V_s + I_L. \tag{20}$$

---

[4]We show how the Kronecker structure of $F^c$ can be exploited to compute $F^c X^l$ in the derivation only.

Consequently, the matrices in equation 18 are defined as $C \in \mathbb{R}^{L \times L}$, $(C^{-1} + V^T V) \in \mathbb{R}^{L \times L}$ and $I_L \in \mathbb{R}^{L \times L}$. In this way, the two operations namely Cholesky decomposition and inversion that are cubic in cost $O(N^3)$ are reduced to the low rank dimension L with complexity $O(L^3)$.

**Derivation:** Firstly, note that sampling from a standard multivariate Gaussian for $X^l$ or $X^s$ is computationally cheap (see equation 17). Given a symmetrical factor for the covariance $\Sigma = F^c F^{c^T}$ (e.g. by Cholesky decomposition), samples can be drawn via $\theta_{\text{MAP}} + F^c X^l$ as depicted in equation 18. Our derivation involves finding such symmetrical factor for the given form of covariance matrix while exploring the Kronecker structure for the sampling computations to bound the complexity as $O(L^3)$.

Let us first reformulate the covariance (inverse of information matrix) as follows.

$$
\begin{aligned}
\Sigma &= \left( (U_{A_{1:a}} \otimes U_{G_{1:g}}) \Lambda_{1:L} (U_{A_{1:a}} \otimes U_{G_{1:g}})^T + D \right)^{-1} \\
&= \left[ D^{\frac{1}{2}} \left( D^{-\frac{1}{2}} (U_{A_{1:a}} \otimes U_{G_{1:g}}) \Lambda_{1:L}^{\frac{1}{2}} \Lambda_{1:L}^{\frac{1}{2}} (U_{A_{1:a}} \otimes U_{G_{1:g}})^T D^{-\frac{1}{2}} + I_{nm} \right) D^{\frac{1}{2}} \right]^{-1} \\
&= D^{-\frac{1}{2}} \left[ \left( (D^{-\frac{1}{2}} (U_{A_{1:a}} \otimes U_{G_{1:g}}) \Lambda_{1:L}^{\frac{1}{2}}) (D^{-\frac{1}{2}} (U_{A_{1:a}} \otimes U_{G_{1:g}}) \Lambda_{1:L}^{\frac{1}{2}})^T + I_{nm} \right) \right]^{-1} D^{-\frac{1}{2}} \\
&= D^{-\frac{1}{2}} \left[ V V^T + I_{nm} \right]^{-1} D^{-\frac{1}{2}}.
\end{aligned}
\tag{21}
$$

Here, we define: $V_s = D^{-\frac{1}{2}} (U_{A_{1:a}} \otimes U_{G_{1:g}}) \Lambda_{1:L}^{\frac{1}{2}}$. Now, a symmetrical factor for $\Sigma = F^c F^{c^T}$ can be found by exploiting the above structure. We let $W^c$ be a symmetrical factor for $V V^T + I_{nm}$ so that $F^c = D^{-\frac{1}{2}} W^{c^{-1}}$ is the symmetrical factor of $\Sigma$. Following the work of Ambikasaran & O'Neil (2014) the symmetrical factor $W^c$ can be found using equations below.

$$
\begin{aligned}
W^c &= I_{nm} + V_s C V_s^T \\
C &= A_c^{-T} (B_c - I_L) A_c^{-1}.
\end{aligned}
\tag{22}
$$

Note that A and B are Cholesky decomposed matrices of $V^T V \in \mathbb{R}^{L \times L}$ and $V^T V + I_L \in \mathbb{R}^{L \times L}$ respectively. As a first result, this operation is bounded by complexity $O(L^3)$ instead of the full parameter dimension N. Now the symmetrical factor for $\Sigma$ can be expressed as follows.

$$
\begin{aligned}
F^c &= D^{-\frac{1}{2}} W^{-1} = D^{-\frac{1}{2}} (I_{nm} + V_s C V_s^T)^{-1} \\
&= D^{-\frac{1}{2}} \left( I_{nm} - V_s (C^{-1} + V_s^T V_s)^{-1} V_s^T \right).
\end{aligned}
\tag{23}
$$

Woodbury's Identity is used here. Now, it follows simply by substitution:

$$
\begin{aligned}
W^l &= \theta_{\text{MAP}} + F^c X^l \text{ where,} \\
F^c &= D^{-\frac{1}{2}} \left( I_{nm} - V_s (C^{-1} + V_s^T V_s)^{-1} V_s^T \right) \\
&= D^{-\frac{1}{2}} \left( I_{nm} - D^{-\frac{1}{2}} (U_{A_{1:a}} \otimes U_{G_{1:g}}) \Lambda_{1:L}^{\frac{1}{2}} (C^{-1} + V_s^T V_s)^{-1} \Lambda_{1:L}^{\frac{1}{2}} (U_{A_{1:a}} \otimes U_{G_{1:g}})^T D^{-\frac{1}{2}} \right).
\end{aligned}
\tag{24}
$$

This completes the derivation of equation 18. As a result, the inversion operation is bounded by complexity $O(L^3)$. Furthermore, the derivation constitutes smaller matrices $U_{A_{1:a}}$ and $U_{G_{1:g}}$ or diagonal matrices $D$ and $I_{mn}$ which can be stored as vectors. In short the complexity has significantly reduced.

Now we further derive computations that exploits rules of Kronecker products. Consider:

$$
F^c X^l = D^{-\frac{1}{2}} \left( I_{nm} - D^{-\frac{1}{2}} (U_{A_{1:a}} \otimes U_{G_{1:g}}) \Lambda_{1:L}^{\frac{1}{2}} (C^{-1} + V_s^T V_s)^{-1} \Lambda_{1:L}^{\frac{1}{2}} (U_{A_{1:a}} \otimes U_{G_{1:g}})^T D^{-\frac{1}{2}} \right).
\tag{25}
$$

Then, it follows by defining inverted matrix $L^c = (C^{-1} + V_s^T V_s)^{-1} \in \mathbb{R}^{L \times L}$ with a cost $O(L^3)$:

$$F^c X^l = D^{-\frac{1}{2}}\Big(I_{nm} - D^{-\frac{1}{2}}(U_{A_{1:a}} \otimes U_{G_{1:g}})\Lambda_{1:L}^{\frac{1}{2}} L^c \Lambda_{1:L}^{\frac{1}{2}}(U_{A_{1:a}} \otimes U_{G_{1:g}})^T D^{-\frac{1}{2}}\Big) X^l. \tag{26}$$

We further reduce this by evaluating $D^{-\frac{1}{2}}$ and defining $X_D^l = D^{-\frac{1}{2}} X^l \in \mathbb{R}^{mn}$ and $P^c = \Lambda_{1:L}^{\frac{1}{2}} L^c \Lambda_{1:L}^{\frac{1}{2}} \in \mathbb{R}^{L \times L}$. We note that this multiplication operation is memory-wise feasible.

$$F^c X^l = X_D^l - \Big(D^{-1}(U_{A_{1:a}} \otimes U_{G_{1:g}}) P^c (U_{A_{1:a}} \otimes U_{G_{1:g}})^T X_D^l\Big). \tag{27}$$

Now, we map $X_D^l$ to matrix normal distribution by an unvec($\cdot$) operation so that $X_D^s = \text{unvec}(X_D^l)$ $\in \mathbb{R}^{n \times n}$ or equivalently $X_D^l = \text{vec}(X_D^s)$. Using a widely known relation for Kronecker product that is - $(U_{A_{1:a}} \otimes U_{G_{1:g}})^T \text{vec}(X_D^s) = \text{vec}(U_{G_{1:g}}^T X_D^s U_{A_{1:a}})$, it follows:

$$F^c X^l = X_D^l - \Big(D^{-1}(U_{A_{1:a}} \otimes U_{G_{1:g}}) P^c \text{vec}(U_{G_{1:g}}^T X_D^s U_{A_{1:a}})\Big). \tag{28}$$

Note that matrix multiplication is performed with small matrices. Repeating a similar procedure as above we obtain the equation below for $X_P^s = P^c \text{vec}((U_{A_{1:a}} \otimes U_{G_{1:g}})^T X_D^l)$,

$$\begin{aligned}
F^c X^l &= X_D^l - \Big(D^{-1}\text{vec}[(U_{A_{1:a}} \otimes U_{G_{1:g}})X_P^s]\Big) \\
&= X_D^l - \Big(D^{-1}\text{vec}(U_{G_{1:g}} X_P^s U_{A_{1:a}}^T)\Big).
\end{aligned} \tag{29}$$

This completes the derivation. Lastly, we provide a remark below to summarize the main points.

**Remark:** We herein presented derivation is to sample from equation 16, a low-rank and information formulation of MND. This analytical solution ensures (a) $O(N^3) >> O(L^3)$ for Cholesky decomposition, (b) $O(N^3) >> O(L^3)$ for a matrix inversion, (c) storage of small matrices $U_{G_{1:g}}$, $U_{A_{1:a}}$, a diagonal matrix $D$ and identity matrices and finally (d) matrix multiplications that only involve these matrices. This is a direct benefit of our proposed LRA that preserves Kronecker structure in eigenvectors.

## C  THEORETICAL ANALYSIS

Some of the interesting theoretical properties are as follows with proofs provided in section D.

### C.1  DIAGONAL CORRECTION LEADS TO MORE ACCURATE ESTIMATION OF INFORMATION MATRIX

A theoretical result of adding a diagonal correction term is captured below. This relates to the work presented in section 2.2 where a diagonal correction term is added to EFB estimates of IM.

**Lemma 1:** *Let $I \in \mathbb{R}^{N \times N}$ be the real Fisher information matrix, and let $I_{def} \in \mathbb{R}^{N \times N}$ and $I_{efb} \in \mathbb{R}^{N \times N}$ be the DEF and EFB estimates of it respectively. It is guaranteed to have $\left\|I - I_{efb}\right\|_F \geq \left\|I - I_{def}\right\|_F$.*

**Corollary 1:** *Let $I_{kfac} \in \mathbb{R}^{N \times N}$ and $I_{def} \in \mathbb{R}^{N \times N}$ be KFAC and our estimates of real Fisher information matrix $I \in \mathbb{R}^{N \times N}$ respectively. Then, it is guaranteed to have $\left\|I - I_{kfac}\right\|_F \geq \left\|I - I_{def}\right\|_F$.*

**Remark:** For interested readers, find the proof $\left\|I - I_{kfac}\right\|_F \geq \|I - I_{efb}\|_F$ in George et al. (2018). Note that $\left\|I - I_{kfac}\right\|_F \geq \|I - I_{efb}\|_F$ may not mean that $\left\|I^{-1} - I_{kfac}^{-1}\right\|_F \geq \left\|I^{-1} - I_{efb}^{-1}\right\|_F$ or vice versa. Yet, our proposed approximation yield better estimates than KFAC in the information form of MND.

### C.2  THEORETICAL PROPERTIES OF LOW-RANK INFORMATION MATRIX

To our knowledge, the proposed sparse IM have not been studied before. Therefore, we theoretically motivate its design and validity for better insights. The analysis can be found below.

Firstly, we study the effects of preserving Kronecker structure in eigenvectors. We define:

$$\hat{I}_{1:K}^{\text{top}} = (U_A \otimes U_G)_{1:K} \Lambda_{1:K} (U_A \otimes U_G)_{1:K}^T \tag{30}$$

as a low rank EFB estimates of true Fisher that preserves top K eigenvalues. Similarly, $\hat{I}_{1:L}^{\text{top}}$ can be defined which preserves top L eigenvalues. In contract, our proposal to preserve Kronecker structure in eigenvectors $\hat{I}_{1:L}$ is denoted as shown below. Now, we provide our analysis with Lemma 2.

$$\hat{I}_{1:L} = (U_{A_{1:a}} \otimes U_{G_{1:g}}) \Lambda_{1:L} (U_{A_{1:a}} \otimes U_{G_{1:g}})^T. \tag{31}$$

**Lemma 2:** *Let $I \in \mathbb{R}^{N \times N}$ be the real Fisher information matrix, and let $\hat{I}_{1:K}^{top} \in \mathbb{R}^{N \times N}$, $\hat{I}_{1:L}^{top} \in \mathbb{R}^{N \times N}$ and $\hat{I}_{1:L} \in \mathbb{R}^{N \times N}$ be the low rank estimates of $I$ of EFB obtained by preserving top K, L and top K plus additional J resulting in L eigenvalues. Here, we define K < L. Then, the approximation error of $\hat{I}_{1:L}$ is bounded as follows: $\left\| I - \hat{I}_{1:L}^{top} \right\|_F \geq \left\| I - \hat{I}_{1:L} \right\|_F \geq \left\| I - \hat{I}_{1:K}^{top} \right\|_F$.*

**Remark:** This bound provides an insight that if preserving top L eigenvalues result in prohibitively too large covariance matrix, our LRA provides an alternative to preserving top K eigenvalues given that K < L. In practise, note that $\hat{I}_{1:L}$ is a memory-wise feasible option as we formulate $\hat{I}_{1:L} = (U_{A_{1:a}} \otimes U_{G_{1:g}}) \Lambda_{1:L} (U_{A_{1:a}} \otimes U_{G_{1:g}})^T$ which preserves the Kronecker structure in eigenvectors. This can be a case where evaluating $(U_{A_{1:a}} \otimes U_{G_{1:g}})$ or $(U_{A_{1:a}} \otimes U_{G_{1:g}})_{1:K}$ is not feasible to store.

**Lemma 3:** *The low rank matrix $\hat{\Sigma} = \left( (U_{A_{1:a}} \otimes U_{G_{1:g}}) \Lambda_{1:L} (U_{A_{1:a}} \otimes U_{G_{1:g}})^T + D \right)^{-1} \in \mathbb{R}^{N \times N}$ is a non-degenerate covariance matrix if the diagonal correction matrix D and LRA $(U_{A_{1:a}} \otimes U_{G_{1:g}}) \Lambda_{1:L} (U_{A_{1:a}} \otimes U_{G_{1:g}})^T$ are both symmetric and positive definite. This condition is satisfied if $(U_{A_{1:a}} \otimes U_{G_{1:g}}) \Lambda_{1:L} (U_{A_{1:a}} \otimes U_{G_{1:g}})_{ii}^T < \mathbb{E}\left[ \delta\theta_i^2 \right]$ for all $i \in \{1, 2, \cdots, d\}$ and with $\Lambda_{1:L} \nsubseteq 0$.*

**Remark:** This Lemma comments on validity of resulting parameter posterior and proves that sparsifying the matrix can lead to a valid non-degenerate covariance if two conditions are met. As non-degenerate covariance can have a uniquely defined inverse, it is important to check these two conditions. We note that searching the rank can be automated with off-line computations that does not involve any data. Thus, it does not introduce significant overhead. In case D does not turn out to be, there are still several techniques that can deal with it. We recommend eigen-value clipping (Chen et al., 2018) or finding nearest positive semi-definite matrices (Higham, 1988). Lastly, $D^{-1}$ does not get numerically unstable when we add a prior precision term and a scaling factor $(ND + \tau I)^{-1}$.

**Lemma 4:** *Let $I \in \mathbb{R}^{N \times N}$ be the real Fisher information matrix, and let $\hat{I}_{def} \in \mathbb{R}^{N \times N}$, $I_{efb} \in \mathbb{R}^{N \times N}$ and $I_{kfac} \in \mathbb{R}^{N \times N}$ be the low rank DEF, EFB and KFAC estimates of it respectively. Then, it is guaranteed to have $\left\| diag(I) - diag(I_{efb}) \right\|_F \geq \left\| diag(I) - diag(\hat{I}_{def}) \right\|_F = 0$ and $\left\| diag(I) - diag(I_{kfac}) \right\|_F \geq \left\| diag(I) - diag(\hat{I}_{def}) \right\|_F = 0$. Furthermore, if the eigenvalues of $\hat{I}_{def}$ contains all non-zero eigenvalues of $I_{def}$, it follows: $\left\| I - I_{efb} \right\|_F \geq \left\| I - \hat{I}_{def} \right\|_F$.*

**Remark:** Lemma 4 shows the optimally in capturing the diagonal variance while indicating that our approach also becomes effective in estimating off-diagonal entries if IM contains many close to zero eigenvalues. Validity of this assumption has been studied by Sagun et al. (2018) where it is shown that the Hessian of overparameterized DNNs tend to have many close-to-zero eigenvalues. Intuitively, from a graphical interpretation of IM, diagonal entries indicate information present in each nodes and off-diagonal entries are links of these nodes (depicted in figure 1). Our sparsification scheme reduces the strength of the weak links (their numerical values) while keeping the diagonal variance exact. This is a result of the diagonal correction after LRA which exploits spectrum sparsity of IM.

## D   PROOFS

### D.1   DIAGONAL CORRECTION LEADS TO MORE ACCURATE ESTIMATION OF INFORMATION MATRIX

**Proposition 1:** *Let $I \in \mathbb{R}^{N \times N}$ be the real Fisher information matrix, and let $I_{def} \in \mathbb{R}^{N \times N}$ and $\hat{I}_{def} \in \mathbb{R}^{N \times N}$ be our estimates of it with rank d and k such that k < d. Their diagonal entries are equal that is $I_{ii} = I_{def_{ii}} = \hat{I}_{def_{ii}}$ for all i = 1, 2, . . . , N.*

*proof:* The proof trivially follows from the definitions of $I \in R^{N \times N}$, $I_{\text{def}} \in R^{N \times N}$ and $\hat{I}_{\text{def}} \in R^{N \times N}$. As the exact Fisher is an expectation on outer products of back-propagated gradients, its diagonal entries equal $I_{ii} = \mathbb{E}\left[\delta\theta_i^2\right]$ for all i = 1, 2, ..., N.

In the case of full ranked $I_{def}$, substituting $D_{ii} = \mathbb{E}\left[\delta\theta_i^2\right] - \sum_{k=1}^{nm}(v_{\alpha,\alpha}\sqrt{\Lambda_k})^2$ with $\sum_{k=1}^{nm}(v_{\alpha,\alpha}\sqrt{\Lambda_k})^2 = (U_A \otimes U_G)\Lambda(U_A \otimes U_G)_{ii}^T$ results in equation 32 for all i = 1, 2, ..., N.

$$
\begin{aligned}
I_{def_{ii}} &= (U_A \otimes U_G)\Lambda(U_A \otimes U_G)_{ii}^T + D_{ii} \\
&= (U_A \otimes U_G)\Lambda(U_A \otimes U_G)_{ii}^T + \mathbb{E}\left[\delta\theta_i^2\right] - (U_A \otimes U_G)\Lambda(U_A \otimes U_G)_{ii}^T = \mathbb{E}\left[\delta\theta_i^2\right]
\end{aligned}
\tag{32}
$$

Similarly, we substitute $\hat{D}_{ii} = \mathbb{E}\left[\delta\theta_i^2\right] - \sum_{k=1}^{NM}(\hat{v}_{\alpha,\alpha}\sqrt{\Lambda_{1:L}})^2$ with $\sum_{k=1}^{NM}(\hat{v}_{\alpha,\alpha}\sqrt{\Lambda_{1:L}})^2 = (U_{A_{1:N}} \otimes U_{G_{1:M}})\Lambda_{1:L}(U_{A_{1:N}} \otimes U_{G_{1:M}})_{ii}^T$ which results in equation 33 for all i = 1, 2, ..., N.

$$
\begin{aligned}
\hat{I}_{def_{ii}} &= (U_{A_{1:N}} \otimes U_{G_{1:M}})\Lambda_{1:L}(U_{A_{1:N}} \otimes U_{G_{1:M}})_{ii}^T + D_{ii} \\
&= (U_{A_{1:N}} \otimes U_{G_{1:M}})\Lambda_{1:L}(U_{A_{1:N}} \otimes U_{G_{1:M}})_{ii}^T + \mathbb{E}\left[\delta\theta_i^2\right] - (U_{A_{1:N}} \otimes U_{G_{1:M}})\Lambda_{1:L}(U_{A_{1:N}} \otimes U_{G_{1:M}})_{ii}^T \\
&= \mathbb{E}\left[\delta\theta_i^2\right]
\end{aligned}
\tag{33}
$$

Therefore, we have $I_{ii} = I_{\text{def}_{ii}} = \hat{I}_{\text{def}_{ii}}$ for all i = 1, 2, ..., N.

**Lemma 1:** *Let $I \in \mathbb{R}^{N \times N}$ be the real Fisher information matrix, and let $I_{def} \in \mathbb{R}^{N \times N}$ and $I_{efb} \in \mathbb{R}^{N \times N}$ be the DEF and EFB estimates of it respectively. It is guaranteed to have $\left\|I - I_{efb}\right\|_F \geq \left\|I - I_{def}\right\|_F$.*

*proof:* Let $e^2 = \|A - B\|_F^2$ define a squared Frobenius norm of error between the two matrices $A \in \mathbb{R}^{N \times N}$ and $B \in \mathbb{R}^{N \times N}$. Now, $e^2$ can be formulated as,

$$
\begin{aligned}
e_b^2 &= \|A - B\|_F^2 \\
&= \sum_i (A - B)_{ii}^2 + \sum_i \sum_{j \neq i} (A - B)_{ij}^2
\end{aligned}
\tag{34}
$$

The first term of equation 34 belongs to errors of diagonal entries in B wrt A whilst the second term is due to the off-diagonal entries.

Now, it follows that,

$$
\begin{aligned}
\|I - I_{\text{efb}}\|_F &\geq \|I - I_{\text{def}}\|_F \\
e_{efb}^2 &\geq e_{def}^2 \\
\sum_i(I - I_{\text{efb}})_{ii}^2 + \sum_i \sum_{j \neq i}(I - I_{\text{efb}})_{ij}^2 &\geq \sum_i(I - I_{\text{def}})_{ii}^2 + \sum_i \sum_{j \neq i}(I - I_{\text{def}})_{ij}^2 \\
\sum_i(I - I_{\text{efb}})_{ii}^2 + \sum_i \sum_{j \neq i}(I - I_{\text{efb}})_{ij}^2 &\geq \sum_i \sum_{j \neq i}(I - I_{\text{def}})_{ij}^2 \\
\sum_i(I - I_{\text{efb}})_{ii}^2 + \sum_i \sum_{j \neq i}(I - I_{\text{efb}})_{ij}^2 &\geq \sum_i \sum_{j \neq i}(I - I_{\text{efb}})_{ij}^2
\end{aligned}
$$

Note that $\sum_i(I - I_{\text{def}})_{ii}^2 = 0$ using proposition 1. Furthermore, $\sum_i \sum_{j \neq i}(I - I_{\text{def}})_{ij}^2 = \sum_i \sum_{j \neq i}(I - I_{\text{efb}})_{ij}^2$ since by definition, $I_{\text{efb}}$ and $I_{\text{def}}$ have the same off-diagonal terms.

**Corollary 1:** *Let $I_{kfac} \in \mathbb{R}^{N \times N}$ and $I_{def} \in \mathbb{R}^{N \times N}$ be KFAC and our estimates of real Fisher Information matrix $I \in \mathbb{R}^{N \times N}$ respectively. Then, it is guaranteed to have $\left\|I - I_{kfac}\right\|_F \geq \left\|I - I_{def}\right\|_F$.*

Find the proof $\left\|I - I_{kfac}\right\|_F \geq \|I - I_{\text{efb}}\|_F$ in George et al. (2018).

## D.2 THEORETICAL PROPERTIES OF LOW-RANK INFORMATION MATRIX

**Lemma 2:** *Let $I \in \mathbb{R}^{N \times N}$ be the real Fisher information matrix, and let $\hat{I}_{1:K}^{top} \in \mathbb{R}^{N \times N}$, $\hat{I}_{1:L}^{top} \in \mathbb{R}^{N \times N}$ and $\hat{I}_{1:L} \in \mathbb{R}^{N \times N}$ be the low rank estimates of $I$ of EFB obtained by preserving top K, L and top K plus additional J resulting in L eigenvalues. Here, we define K < L. Then, the approximation error of $\hat{I}_{1:L}$ is bounded as follows: $\left\|I - \hat{I}_{1:L}^{top}\right\|_F \geq \left\|I - \hat{I}_{1:L}\right\|_F \geq \left\|I - \hat{I}_{1:K}^{top}\right\|_F$.*

*proof:* From the definition, $(U_A \otimes U_G)\Lambda(U_A \otimes U_G)^T = V\Lambda V^T$ is PSD as $\Lambda_{ii} = \mathbb{E}\left[(V^T \delta\theta)_i^2\right] \geq 0$ for all elements i and $VV^T = I$ with $I$ as an identity matrix (orthogonality). Naturally, low rank approximations $(U_A \otimes U_G)_{1:L^{\mathrm{top}}}\Lambda_{1:L^{\mathrm{top}}}(U_A \otimes U_G)_{1:L^{\mathrm{top}}}^T$, $(U_A \otimes U_G)_{1:K^{\mathrm{top}}}\Lambda_{1:K^{\mathrm{top}}}(U_A \otimes U_G)_{1:K^{\mathrm{top}}}^T$ and $(U_{A_{1:a}} \otimes U_{G_{1:g}})\Lambda_{1:L}(U_{A_{1:a}} \otimes U_{G_{1:g}})^T = (U_A \otimes U_G)_{1:L}\Lambda_{1:L}(U_A \otimes U_G)_{1:L}^T$ are again PSD by the fact that low rank approximation does not introduce negative eigenvalues.

Now, a well known fact from dimensional reduction literature is that low rank approximation preserving the top eigenvalues result in best approximation errors in terms of Frobenius norm for the given rank. Informally stating Wely's ideas on eigenvalue perturbation:

Let $B \in \mathbb{R}^{m \times n}$ with rank smaller or equal to p (one can also use complex space $\mathbb{C}$ instead of $\mathbb{R}$) and let $E = A - B$ with $A \in \mathbb{R}^{m \times n}$. Then, it follows that,

$$\|A - B\|_F^2 = \sigma_1(A-B)^2 + \cdots + \sigma_\mu(A-B)^2 \geq \sigma_{p+1}(A-B)^2 + \cdots + \sigma_\mu(A-B)^2 = \|A - B_{1:p}\|_F^2, \quad (35)$$

where $\sigma_1, \cdots \sigma_\mu$ are the singular values of A with $\mu = \min(n, m)$. The convention here is that $\sigma_i(A)$ is the ith largest singular value and $\sigma_i(A) = 0$ for $i > \mathrm{rank}(A)$. Using this insight, and the fact that in the given settings, squared singular values are variances in new space lead to:

$$\left\|I - \hat{I}_{1:K}^{\mathrm{top}}\right\|_F \geq \left\|I - \hat{I}_{1:L}\right\|_F \geq \left\|I - \hat{I}_{1:L}^{\mathrm{top}}\right\|_F$$

**Lemma 3:** *The low rank matrix $\hat{\Sigma} = \left((U_{A_{1:a}} \otimes U_{G_{1:g}})\Lambda_{1:L}(U_{A_{1:a}} \otimes U_{G_{1:g}})^T + D\right)^{-1} \in \mathbb{R}^{N \times N}$ is a non-degenerate covariance matrix if the diagonal correction matrix D and LRA $(U_{A_{1:a}} \otimes U_{G_{1:g}})\Lambda_{1:L}(U_{A_{1:a}} \otimes U_{G_{1:g}})^T$ are both symmetric and positive definite. This condition is satisfied if $(U_{A_{1:a}} \otimes U_{G_{1:g}})\Lambda_{1:L}(U_{A_{1:a}} \otimes U_{G_{1:g}})_{ii}^T < \mathbb{E}\left[\delta\theta_i^2\right]$ for all $i \in \{1, 2, \cdots, N\}$ and with $\Lambda_{1:L} \nsubseteq 0$.*

*proof:* Let us first rewrite $\hat{I}_{\mathrm{def}} = (U_{A_{1:a}} \otimes U_{G_{1:g}})\Lambda_{1:L}(U_{A_{1:a}} \otimes U_{G_{1:g}})^T + D$ in the following form.

$$\begin{aligned}
(U_{A_{1:a}} \otimes U_{G_{1:g}})\Lambda_{1:L}(U_{A_{1:a}} \otimes U_{G_{1:g}})^T + D &= (U_{A_{1:a}} \otimes U_{G_{1:g}})\Lambda_{1:L}^{\frac{1}{2}}\Lambda_{1:L}^{\frac{1}{2}}(U_{A_{1:a}} \otimes U_{G_{1:g}})^T + D \\
&= \left[(U_{A_{1:a}} \otimes U_{G_{1:g}})\Lambda_{1:L}^{\frac{1}{2}}\right]\left[(U_{A_{1:a}} \otimes U_{G_{1:g}})\Lambda_{1:L}^{\frac{1}{2}}\right]^T + D \quad (36) \\
&= UU^T + D
\end{aligned}$$

Now, if $D$ and $(U_{A_{1:a}} \otimes U_{G_{1:g}})\Lambda_{1:L}(U_{A_{1:a}} \otimes U_{G_{1:g}})^T$ is both symmetric and positive definite, it follows that for an arbitrary vector $x \in \mathbb{R}^d$, $x^T UU^T x > 0$ as eigen-values $R_i > 0$ by construction. Furthermore, $x^T Dx > 0$ also holds by the definition of positive definiteness. Therefore, we have $x^T(UU^T + D)x = x^T UU^T x + x^T Dx > 0$ which leads to the proof that $I_{\mathrm{def}}$ is positive definite if $D$ and $(U_{A_{1:a}} \otimes U_{G_{1:g}})\Lambda_{1:L}(U_{A_{1:a}} \otimes U_{G_{1:g}})^T$ is both symmetric and positive definite. As this results in non-degenerate IM, the canonical covariance $\Sigma$ is non-degenerate as well.

Trivially following the definition of $D_{ii} = \mathbb{E}\left[\delta\theta_i^2\right] - (U_A \otimes U_G)\Lambda(U_A \otimes U_G)_{ii}^T$, $D_{ii} > 0$ for all i when $(U_{A_{1:a}} \otimes U_{G_{1:g}})\Lambda_{1:L}(U_{A_{1:a}} \otimes U_{G_{1:g}})_{ii}^T < \mathbb{E}\left[\delta\theta_i^2\right]$. Again, by the definition of $\Lambda_{ii} = \mathbb{E}\left[(V^T \delta\theta)_i^2\right] \geq 0$, $\Lambda_{1:L}$ containing no zero eigenvalues result in the positive definite matrix $(U_{A_{1:a}} \otimes U_{G_{1:g}})\Lambda_{1:L}(U_{A_{1:a}} \otimes U_{G_{1:g}})^T$.

**Lemma 4:** *Let $I \in \mathbb{R}^{N \times N}$ be the real Fisher information matrix, and let $\hat{I}_{def} \in \mathbb{R}^{N \times N}$, $I_{efb} \in \mathbb{R}^{N \times N}$ and $I_{kfac} \in \mathbb{R}^{N \times N}$ be the low rank DEF, EFB and KFAC estimates of it respectively. Then, it is guaranteed to have $\left\|diag(I) - diag(I_{efb})\right\|_F \geq \left\|diag(I) - diag(\hat{I}_{def})\right\|_F = 0$ and $\left\|diag(I) - diag(I_{kfac})\right\|_F \geq \left\|diag(I) - diag(\hat{I}_{def})\right\|_F = 0$. Furthermore, if the eigenvalues of $\hat{I}_{def}$ contains all non-zero eigenvalues of $I_{def}$, it follows: $\left\|I - I_{efb}\right\|_F \geq \left\|I - \hat{I}_{def}\right\|_F$.*

*proof:* The first part follows from proposition 1 which states that for all the elements i, $I_{ii} = \hat{I}_{\mathrm{def}}$, $\left\|diag(I) - diag(I_{\mathrm{efb}})\right\|_F \geq \left\|diag(I) - diag(\hat{I}_{\mathrm{def}})\right\|_F = 0$ and $\left\|diag(I) - diag(I_{\mathrm{kfac}})\right\|_F \geq \left\|diag(I) - diag(\hat{I}_{\mathrm{def}})\right\|_F = 0$. This results by the design of the method, in which, we correct the diagonal entries in parameter space after the LRA.

For the second part of the proof, lets recap that Lemma 2 (Wely's idea on eigenvalue perturbation) that removing zero eigenvalues does not affect the approximation error in terms of Frobenius norm. This then implies that off-diagonal elements of $\hat{I}_{\text{def}}$ and $I_{\text{efb}}$ are equivalent. Then,:

$$\|I - I_{\text{efb}}\|_F \geq \left\|I - \hat{I}_{\text{def}}\right\|_F$$
$$e_{efb}^2 \geq e_{def}^2$$
$$\sum_i (I - I_{\text{efb}})_{ii}^2 + \sum_i \sum_{j \neq i} (I - I_{\text{efb}})_{ij}^2 \geq \sum_i (I - \hat{I}_{\text{def}})_{ii}^2 + \sum_i \sum_{j \neq i} (I - \hat{I}_{\text{def}})_{ij}^2$$
$$\sum_i (I - I_{\text{efb}})_{ii}^2 + \sum_i \sum_{j \neq i} (I - I_{\text{efb}})_{ij}^2 \geq \sum_i \sum_{j \neq i} (I - \hat{I}_{\text{def}})_{ij}^2$$
$$\sum_i (I - I_{\text{efb}})_{ii}^2 + \sum_i \sum_{j \neq i} (I - I_{\text{efb}})_{ij}^2 \geq \sum_i \sum_{j \neq i} (I - I_{\text{efb}})_{ij}^2$$

Again, $\sum_i (I - \hat{I}_{\text{def}})_{ii}^2 = 0$ according to proposition 1 for all the elements i.

## E    IMPLEMENTATION DETAILS AND FURTHER RESULTS

KFAC library from Tensorflow [5] was used to implement the Fisher estimator (Martens & Grosse, 2015) for our methods and the works of Ritter et al. (2018a). Note that empirical Fisher usually is not a good estimates as it is typically biased (Martens & Grosse, 2015) and therefore, we did not use it. KFAC library offers several estimation modes for both fully connected and convolutional layers. We have used the gradients mode for KFAC Fisher estimation (which is also crucial for our pipelines) whereas the exact mode was used for diagonal approximations. We did not use the exponential averaging for all our experiments as well as the inversion scheme in the library. However, when using it in practice, it might be useful especially if there are too many layers that one cannot access convergence of the Fisher estimation. We have used NVIDIA Tesla for grid searching the parameters of Diag and KFAC Laplace, and 1080Ti for all other experiments.

### E.1    TOY REGRESSION DATASET

Apart from the architecture choices discussed in section 4, the training details are as follows. A gradient descent optimizer from tensorflow has been used with a learning rate of 0.001 with zero prior precision or L2 regularization coefficient ($\tau = 0.2$ for KFAC, $\tau = 0.45$ for Diag, $N = 1$ and $\tau = 0$ for both FB and DEF have been used). Mean squared error (MSE) has been used as its loss function. Interestingly, the exact block-wise Hessian and their approximations for the given experimental setup contained zero values on its diagonals. This can be interpreted as zero variance in information matrix, meaning no information, resulting in information matrix being degenerate for the likelihood term. In such cases, the covariance may not be uniquely defined (Thrun et al., 2004). Therefore, we treated these variances deterministic, making the information matrix non-degenerate (motivated from Lemma 3). Similar findings are interestingly reported by MacKay (1992).

More importantly, we present a detailed analysis to avoid misunderstanding about our toy dataset experiments. As a starting remark, a main advantage of this toy regression problem is that it simplifies the understandings of on-going process, in lieu of sophisticated networks with a large number of parameters. Typically, as of Herandez-Lobato & Adams (2015), Ritter et al. (2018a), Gal (2016), or even originating back to Gaussian processes literature, this example has been used to check the predictive uncertainty by qualitatively evaluating on whether the method predicts high uncertainty in the regimes of no training data. However, a drawback exists: no quantitative analysis has been reported to our knowledge other than qualitatively comparing it to community wide accepted ground truth such as Hamiltonian Monte Carlo Sampling (Neal, 1996), and LA using KFAC and Diag seem to be sensitive to hyperparameters in this dataset which makes the comparison difficult.

This is illustrated in figure 7 where we additional introduce Random which is just a user-set $\tau I$ for covariance estimation in order to demonstrate this. Qualitatively analyzing from the first look, all the methods look very similar in delivering high uncertainty estimates in the regimes of no training data. Here, we note that the same hyperparameter settings have been used for Diag, KFAC and FB Laplace whereas the user-set $\tau = 7$ has been found for Random. This agrees to the discussions of Ritter et al. (2018a) as KFAC resulted in less $\tau$ when compared to Diag Laplace.

---

[5]https://github.com/tensorflow/kfac

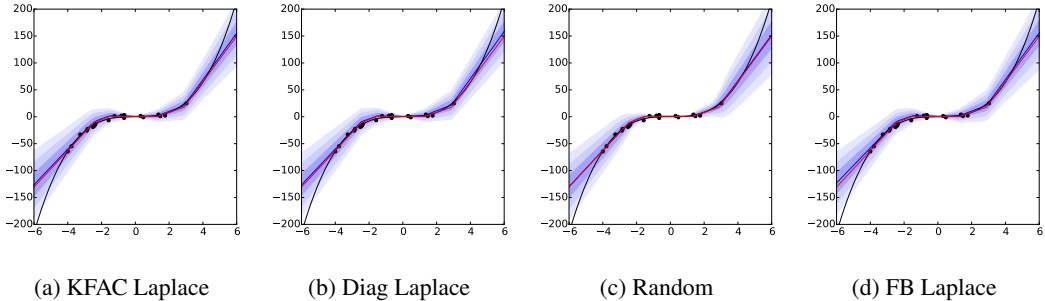

(a) KFAC Laplace     (b) Diag Laplace     (c) Random     (d) FB Laplace

Figure 7: **Toy regression uncertainty.** User-set Laplace means user-set $\tau I$ for covariance estimation. This shows that if one tries to tune the "regularizing" parameters, all these approximations to the true Hessian behaves similarly within this experiment. Now trained with 20 data points.

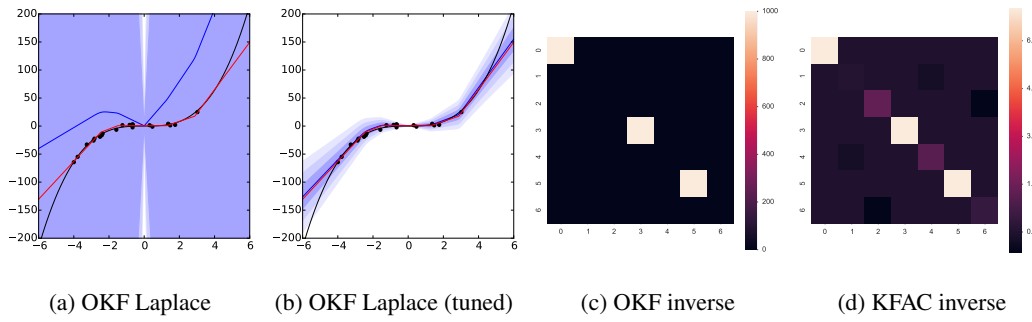

(a) OKF Laplace     (b) OKF Laplace (tuned)     (c) OKF inverse     (d) KFAC inverse

Figure 8: **Toy regression uncertainty and covariance visualization** (only the first layer is shown here). OKF Laplace means using the left hand side of equation 3 without further approximation (only possible with this small model and data-set).

However, we also observed that without the approximation step of equation 3 (denoted OKF), using the same hyper parameter as above resulted in visible over-prediction of uncertainty and inaccurate estimates on the prediction. This is shown in figure 8. Again, tuning the parameter to a higher precision $\tau$, similar behavior to figure 7 can be reproduced. This can be analyzed by visualizing the covariance of KFAC and OKF. As it can be seen, in this experiment settings, figure 8 shows that equation 3 damps the magnitude of estimated covariance matrix.

A possible explanation is that if the approximate Hessian is degenerate, then small $\tau I$ places a big mass on areas of low posterior probabilities for some network parameters with no information (zero variance and correlations in the approximate Hessian). This can be seen in figure 8 part (a) where the approximate Hessian contains 3 parameters with exactly zero diagonal elements and zeros in its off-diagonal elements. If one tries to add a small $\tau = 0.001$ here, then the covariance of these parameters get close to its inverse $\tau^{-1} = 1000$ as shown in figure 8 part (c). This would in return result in over prediction of uncertainty and inaccurate predictions which explains figure 8 part (a).

Another interesting experiments are studying the effects of dataset size to number of parameters. For this, we have increased the dataset size to 100 in oppose to 20. Again, we now compare the approximate Hessian by visualizing them. Notably, at using 100 data points resulted in more number of zero diagonal entries and corresponding rows and columns. This is due to over parameterization of the model which results in under determined Hessian.

These insights hint for the followings. Accurately estimating the Hessian while forcing its estimates non-degeneracy via not considering zero eigenvalues for this data and model can lead to less sensitivity to its hyperparameters or $\tau$ in particular. Secondly, further increasing or decreasing the ratio of data points to number of parameters change the approximate Hessian (similarly found for estimates of Fisher) changes its structure, and can lead to under-determined approximation (therefore, changing its loss landscape). Finally, if the Hessian is under-determined, hyperparameters $\tau$ affects the resulting predictive uncertainty (or covariance) if its magnitude significantly differs (and in case of

KFAC). However, as more detailed experimental analysis is outside the scope of the paper, can be an interesting future work to further analyze the relation between the hyperparameters, their probabilistic interpretation and resulting loss landscape of neural network.

### E.2 BENCHMARK IMPLEMENTATIONS

We have used Numpyro (Phan et al., 2019) for the implementations of HMC. We have used 50000 MC samples to generate the results in order to ensure the convergence. For the implementation of Bayes By Backprop we have used an open-source implementation `https://github.com/ThirstyScholar/bayes-by-backprop` for which a similar experiment settings are implemented where the Gaussian noise is sampled in a batch initially, and a symmetric sampling technique is deployed. We note that the number of data samples and network architectures are different. Furthermore, we have used 10000 iterations to ensure convergence of the network.

### E.3 CLASSIFICATION TASKS

Most of the implementations for MNIST and CIFAR10 experiments were taken from Tensorflow tutorials [6] including the network architectures and training pipelines if otherwise stated in the main text. This is in line of argument that our method can be directly applied to existing, and well trained neural networks. For MNIST experiments, the architecture choices are the followings. Firstly, no down-scaling has been performed to its inputs. The architecture constitutes 2 convolutional layers followed by 2 fully connected layer (each convolutional layer is followed by a pooling layer of size 2 by 2, and a stride 2). For flattening from the second convolutional layer to the first fully connected layer, a pooling operation of 49 by 64 has been naturally used. RELU activation have been used for all the layers except the last layer which computes the softmax output. Dropout has been applied to the fully connected layer with dropout rate of 0.6 after a grid search (explained in section E.3.1). Regarding the loss functions, cross entropy loss has been used with ADAM as its optimizer and learning rate of 0.001. An important information is the size of each layers. The first layer constitutes 32 filters with 5 by 5 kernel, followed by the second layer with 64 filters and 5 by 5 kernel. The first fully connected layer then constitutes 1024 units and the last one ends with 10 units. We note that, this validates our method on memory efficiency as the third layer has a large number of parameters, and its covariance, being quadratic in its size, cannot be stored in our utilized GPUs.

Regarding the architecture selection of CIFAR10 experiments, no down-scaling of the inputs has be done. The chosen architecture is composed of 2 convolutional layers followed by 3 fully connected layers. Pooling layers of size 3 by 3 with strides 2 have been applied to outputs of the convolutional layers. Obviously, the third convolutional layer is pooled to match the input size of the following fully connected layers. Batch normalization has been applied to each outputs of convolutional layer before pooling, with bias 1, $\alpha$ of 0.001/9.0 and $\beta$ of 0.75 (notations are different to the main text, and this follows that of tensorflow library). A weight decay factor of 0.004 has been used, and trained again with cross entropy loss, now with

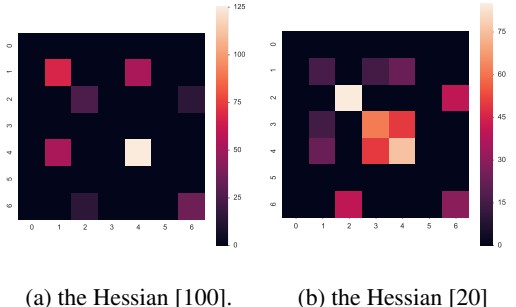

(a) the Hessian [100].  (b) the Hessian [20]

Figure 9: **Visualization** of the approximate Hessian with different data points.

a stochastic gradient descent. Learning rate of 0.001 has been used. Again, the most relevant settings are: the first layer constitutes 5 by 5 kernel with 64 filters. This is then again followed by the same (but as input to CIFAR10 is RGB, the second layer naturally has more number of parameters). Units of 384, 192, and 10 have been used for the fully connected layers in an ascending order. Lastly, random cropping, flipping, brightness changes and contract have been applied as the data augmentation scheme. Similar to MNIST experiments, the necessity of LRA is capture in CIFAR10 as well.

---

[6]`https://www.tensorflow.org/tutorials`

Unlike Ritter et al. (2018a) we did not artificially augment the data for MNIST experiments because the usual training pipeline did not require it. For our low rank approximation, we always have used the maximum rank we could fit, after removing all the zero eigenvalues and checking the conditions from *Lemma 3*. Lastly we have used 1000 Monte-Carlo samples for MNIST, and 100 samples for CIFAR10 and toy regression experiments.

### E.3.1 BENCHMARK IMPLEMENTATIONS

Implementation of deep ensemble (Lakshminarayanan et al., 2017) was kept rather simple by not using the adversarial training, but we combined 15 networks that were trained with different initialization. The same architecture and training procedure were used for all. Note that CIFAR10 experiments with similar convolutional architectures were not present in the works of (Lakshminarayanan et al., 2017) to the best of our knowledge. On MNIST, Louizos & Welling (2017) found similar results to ours that deep ensemble performed similar to the MC-dropout (Gal, 2016). For dropout, we have tried a grid search of dropout probabilities of 0.5 and 0.8, and have reported the best results. For the methods based on Laplace approximation, we have performed grid search on hyperparameters N of (1, 50000, 100000) and 100 values of $\tau$ were tried using known class validation set. Note that for every method, and different data-sets, each method required different values of $\tau I$ to give a reasonable accuracy. The starting point of the grid-search were determined based on if the mean values of their predictions were obtained similar accuracy to the deterministic counter parts. The figure below are the examples on MNIST where minimum ece points were selected and reported.

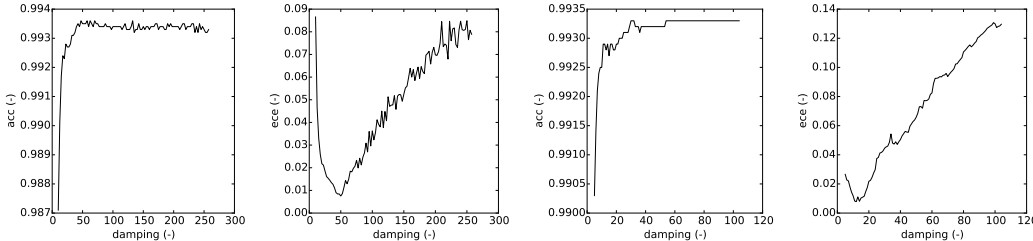

Figure 10: **Grid search results.** For Diag (left two figures) and KFAC Laplace (right two) an extensive grid search has been conducted to ensure fair comparison. Here, we report the results with pseudo observation term of 50000 on MNIST. This ensures that a main difference to DEF Laplace is the expression for model uncertainty as the inference and network architectures are kept the same.

