# OpenReview forum: "Representing Model Uncertainty of Neural Networks in Sparse Information Form"
_ICLR.cc/2020/Conference — Reject_

### Official Review · AnonReviewer2 · 2019-10-19
**Official Blind Review #2**

**Rating:** 1

**Review:**

The contribution of the paper is marginal, as the principle of imposing Gaussians on the network to perform Bayes is not new. The Laplace-based approximation is half-baked and certainly much better techniques for Bayes exist in the recent literature; furthermore, it's selection is not substantiated enough, and, of course, it represents no novelty. The experimental results are not convincing, as both the considered scenarios are limited and the comparisons are too poor (no consideration of state of the art alternatives). The provided corollaries are not actually helpful and should be put in the appendix.

**Experience Assessment:**

I have published in this field for several years.

**Review Assessment: Checking Correctness Of Derivations And Theory:**

I carefully checked the derivations and theory.

**Review Assessment: Checking Correctness Of Experiments:**

I carefully checked the experiments.

**Review Assessment: Thoroughness In Paper Reading:**

I read the paper thoroughly.

---

> ### Author Response · Authors · 2019-11-11
> **We hope for further discussions.**
>
> Thank you for the time you spent reviewing our paper. Here, we express our concerns about strong claims you have made.
>
> You claim: "The contribution of the paper is marginal, as the principle of imposing Gaussian on the network to perform Bayes is not new."
>
> - We strongly disagree that our contributions are marginal. Despite the principle of imposing Gaussian on the networks is pioneered by MacKay 1992, the method of imposing Gaussian for deep neural networks, without resorting to fully factorized or matrix normal distributions, is certainly new. We achieve this in information formulation of Gaussian, exploiting the Kronecker structure and natural spectral sparsity in information matrix (inverse of the covariance matrix). We have made several bullet points summarizing the main contributions in our introduction, which we hope you consider them in the review.
>
> You claim: "The Laplace-based approximation is half-baked and certainly much better techniques for Bayes in the literature;".
>
> - Please support your claim here. What aspects of Laplace-based approximation is half-baked? Which papers are you referring to? If there are better techniques in recent literature, what are they and in what aspect? We hope to hear from you.
>
> You claim: "its selection is not substantiated enough, and, of course, it represents no novelty".
>
> - We argue why Laplace-based methods are relevant for Bayesian Deep Learning and beyond.
>
> 1. Laplace approximation is easy-to-use in the sense that it does not require changes in the training procedure. Practitioners can transform their network architectures to Bayesian by simply computing the approximate Hessian. Since each application domain often have "well working but sensitive to changes" architectures and training scheme, approximate Bayesian inference tools such as Laplace approximation can be useful in practice.
>
> 2. Laplace approximation is scalable to deep neural networks due to several progresses in scalable approximation of the Hessian from 2nd order optimization community. KFAC is a very convincing example: it is scalable to even ImageNet networks and well built software infrastructure already exists.
>
> 3. Core of Laplace approximation is scalable approximation of the Hessian. Improving how we approximate the Hessian of deep neural networks has impact beyond Bayesian Deep Learning. Continual learning, 2nd order optimization and network compression are just few examples.
>
> We also point out that your claim on "no novelty" is not communicated well. The contributions we present have not been done before, and therefore, they contain novelty.
>
> You claim: "The experimental results are not convincing, as both the considered scenarios are limited and the comparisons are too poor (no consideration of state of the art alternatives)."
>
> - Our experiments show the benefits of our approach and the comparisons to the state-of-the-art alternatives are provided.
>
> 1. The considered scenarios are carefully controlled experiments that show case the benefits of our approach. The toy regression examples show (1) the quality of uncertainty estimation and (2) the quality of the Fisher approximations. Our classification experiments show (1) the quality of uncertainty estimation in a more realistic dataset, and (2) the necessity of our low-rank approximation. Please support your claim stating why the considered scenarios are limited.
>
> 2. We compare to the state-of-the art alternatives that are both scalable and training-free. Amongst this class of methods, the presented baselines are strong alternatives and often applied in practice. Please let us know what you exactly meant by "the state-of-the-art alternatives".
>
> You comment: "The provided corollaries are not actually helpful and should be put in the appendix."
>
> - We have incorporated this suggestion and all the theoretical analysis are in the appendix for the current revision. Thank you for your valuable opinion on this.
>
> As a concluding remark we sincerely hope for further discussions.

---

### Official Review · AnonReviewer4 · 2019-10-31
**Official Blind Review #4**

**Rating:** 3

**Review:**

Laplace approximation has been an important tool for obtaining uncertainty estimation for deterministic models. To efficiently approximate the Hessian matrix for neural networks, Ritter et. al (2018) proposes to use K-FAC . Motivated by the relatively inaccurate approximations of K-FAC, this paper proposes to improve KFAC approximations by combining eigen-basis corrections (EK-FAC, George et al, 2018) and diagonal corrections. The paper shows that the proposed method has smaller Frobenius approximate errors compared to K-FAC and EK-FAC. To further reduce the computational costs, the paper proposes a low-rank approximation by keeping only the L largest eigenvalues. Empirically, the paper demonstrates improved calibration and out-of-distribution entropies compared to previous approaches.

# Diagonal Corrections.
With the diagonal correction approximating the Fisher better, the paper shows the computation can still be conducted in the scale of W, which is similar to K-FAC. However, the method requires the diagonal correction matrix D is always positive, which might not be true. Moreover, because the computation requires D^{-1}, clipping D to a small constant will bring up stability issues. I wonder how this problem is tackled in this paper.

# Writing
The paper's notations are messy, which requires a lot of intellectual guesses to understand the conveyed idea.
1) Notations are not introduced, such as $\delta \theta$, $W_{map}^{IV}$, the MN distribution in the appendix.
2) Notations are typoed. Eq(3) $N(0, A^{-1} \otimes G^{-1}) = MN(0, G^{-1}, A^{-1})$; The bottom paragraph in Page 4; eq(8).
3) Notations are abused. In particular for $V$ and $\Lambda$ when introducing EK-FAC. The paper uses $V$ for both true eigenbasis and the EK-FAC eigenbasis. In addition, the EK-FAC part should be moved to the background section.
4) The paragraph below Corollary 1 says the author can prove the proposed method also has closer approximations in terms of the Fisher inverse. But no proofs are given.
5) Caption of Figure4.

# Laplace Approximation
For eq(4, 8, 12), The Hessian in Laplace approximations should be divided by $N$. Although the paper also mentions the scaling below those equations. But technically eq(4, 8, 12) are wrong and I don't know whether the experiments really did the scaling or not.

# Low-rank Approximation
1) It is not clear why the low-rank approximation is necessary. The computational costs are inevitable to compute the eigen-system of A and G. Why do we need the low-rank approximation after that ? K-FAC is not a computationally expensive method either.
2) I cannot understand the proofs of Lemma 2. In fact, I don't know what $ I_{1:L}^{top}, I_{1:K}^{top}$ means. More explicit formulas should be given for clarity.
3) Lemma 4 states $I_ii = (\hat{I}_{def})_ii$. Although the diagonal correction makes $I_ii = (I_{def})_ii$, the low-rank approximation makes them unequal again. Or I guess you use a different D in eq(13) from the D in eq(10) ?

# Experiments
The paper needs more experiments to validate the proposed method. Firstly, for the MNIST experiments, it is better to use the same architecture as in Ritter et al (2018) for direct comparisons. Beyond that, the adversarial attack experiment and the mis-classification uncertainty experiment in Ritter et al (2018) seem to be good choices as well.

# Overall
The paper proposes a diagonal corrected EK-FAC, achieving better approximation of the Fisher matrix. Interestingly, the paper shows that this corrections doesn't add too much computations.  However, the proposed low-rank approximation doesn't seem necessary. And the paper's notations and presentations are too messy to be an accepted paper.

**Experience Assessment:**

I have published one or two papers in this area.

**Review Assessment: Checking Correctness Of Derivations And Theory:**

I carefully checked the derivations and theory.

**Review Assessment: Checking Correctness Of Experiments:**

I carefully checked the experiments.

**Review Assessment: Thoroughness In Paper Reading:**

I read the paper thoroughly.

---

> ### Author Response · Authors · 2019-11-11
> **Thank you for very useful comments.**
>
> Thank you for very useful comments, and your review has improved our paper significantly. We start by addressing your main points.
>
> You comment: "However, the proposed low-rank approximation doesn't seem necessary."
>
> - The proposed low-rank approximation is strictly necessary. Let us illustrated with an example. If a layer of the network has 1 million parameters, the covariance matrix of Multivariate Normal Distribution is a matrix with size 1 million by 1 million which is intractable to sample or even store on standard computers. This is a main reason why many approaches resort to a diagonal approximation (then the covariance is a vector of 1 million) or a matrix normal distribution (covariance is decomposed into Kronecker products of two smaller matrices). However, we find these two approximations not convincing in their expressiveness; diagonal approximations ignore correlations between the parameters and matrix normal distribution are not accurate on their variances (which arguably is very important). Our main research question was therefore on proposing alternatives to these two.
>
> Now, adding a diagonal correction leads to information formulation of Multivariate Normal Distribution, and if its information matrix is of size 1 million by 1 million as an example, it is computationally intractable to sample from it. Thankfully, the eigenvalues of information matrix tends to be very sparse and we can explore this insight in developing a sampler that is cubic in cost for the chosen dimension of lower rank L. In a full ranked case where L equals N (1 million), then it is computationally not feasible to invert and sample. If you ask yourself how you can sample from equation 6 (in the new revision), this aspect will become more clear. We have added figure 1 and example 1 in the new revision, clarifying your concern that the proposed low-rank approximation may not be necessary. Furthermore, we have made the section on derivation of sampling computations clearer.
>
> You comment: "And the paper's notations and presentations are too messy to be an accepted paper."
>
> - We sincerely apologize if you had hard time reading our paper. We have thoroughly revised to improve the clarity of the paper in the new revision. Overall, we believe the paper is more accessible and clearer (thanks to your review!).
>
> Your other points: Now, we address the other points you have concerns on.
>
> On "Diagonal Corrections":
>
> - 1. As explained in Lemma 3, the matrix D requires to be always positive and this is indeed, not always true. However, with low-rank approximation and adding the diagonal correction afterwards, you can ensure that D remains strictly positive. Furthermore, eigenvalue clipping or finding closest positive definite matrices are other techniques that can be employed (which appear quite often in 2nd order optimization communities). We have explained this point in the new revision.
>
> 2. The inversion of matrix D is stable because of the prior precision term and scaled by the number of data points (see equation 9). Our derivation of the analytical sampler works on equation 9 which we clarify in the new revision (we omitted these term for achieving better clarity).
>
> On "Writing":
>
> - We apologize if you had to guess a lot. We have improved the clarity of the paper significantly.
>
> 1. All the notations are introduced in the new revision. We apologize for our careless mistakes.
>
> 2. All the typing mistakes are corrected in the new revision. We apologize for our careless mistakes.
>
> 3. We have incorporated your suggestions on notations for EK-FAC. Unfortunately, we do not move EK-FAC part to background section to keep the story-line the same (R5 finds the structure clear). Instead, we clearly point out what parts of the paper contains our contributions in the introduction (in the summary of our contributions) and shortened the parts on EK-FAC. Please let us know if this is unsatisfactory.
>
> 4. We did not claim that our proposed method also has closer approximations in terms of the Fisher inverse and therefore, no proofs are given. Please check again the paragraph below Corollary 1.
>
> 5. We apologize for our careless mistakes and we have made significant efforts to improve the clarity of the paper in the new revision.

---

> > ### Author Response · Authors · 2019-11-11
> > **Continued.**
> >
> >
> > On "Laplace Approximation":
> >
> > - In fact, the Fisher should be scaled by N and therefore, these are technically correct. We refer to references [1,2] for technical details. We have also briefly noted on this point for the current revision.
> >
> > On "Low-rank Approximation":
> >
> > - We address each points of your concerns below.
> >
> > 1. Please find the explanation above on why low-rank approximation is necessary. It is not to compute the eigen-system of A and G. It is to sample from the resulting distribution from its information formulation (eq. 6). Indeed K-FAC is not a computationally expensive method either but diagonal correction to the eigenbasis of K-FAC results in an Multivariate Normal Distribution whereas K-FAC results in matrix normal distribution. Please compare eq. 3 and eq. 6 and try to sample from eq. 6. You will see that it is non-trivial without the low-rank approximation. We have added a figure, an example and a table to address this point in the new revision.
> >
> > 2. We apologize for this, and we have added explicit formulas. Thank you for the good suggestion.
> >
> > 3. Diagonal correction is added after the low rank approximation as illustrated in algorithm 2 of old version. The mathematics of computing the diagonal term is the same and we made algorithm 2 in order to provide an overview. Now, we also show this with a figure (figure 1) and comment in the footnote of equation 8 (new revision). Thank you for pointing this out and it has improved the clarity of the paper.
> >
> > On "Experiments":
> >
> > - Our experiments validates the proposed method and are carefully designed to show the benefits of our approach.
> >
> > The choice of architectures are to ensure that the given low-rank approximation is necessary. In fact, [4] uses fully connected layers with small sizes only and for this, one needs no low-rank approximation (with the architecture of [4] we cannot say we validated our method). On the other hand, the 3rd layer of our architecture on MNIST contains significantly more large number of parameters where the low rank approximation is strictly necessary (as a representative for scalability). We hope that this point is automatically answered when you understand why the low rank approximation is necessary.
> >
> > Furthermore, we choose to evaluate on the Fisher estimates instead of the adversarial examples, and the mis-classification uncertainty for in-domain sets are already covered in our experiments. One novel part in our experiments is that we evaluate both in-domain and out-domain sets with the same hyperparameter choices. This is because we found that [2]'s way of evaluating can be misleading: one can choose the hyperparameter so that it performs well for out-of-distribution sets but do not generalize to the other metrics that is calibration and accuracy for in-domain datasets. However, comparing on direction evaluations on the Fisher estimates on MNIST with smaller architectures seem good idea and we will try to include them before the revision period ends. (@update: instead of this experiment we have focused on your updated review and so, for the toy regression experiments, we added an ablation study where we lower the ranks of DEF and also show EFB results. Please find the discussion part of section 4.1 in the new revision, and please let us know if you this concerns you.)
> >
> > Lastly, we have also made significant efforts of grid searching hyperparameters in order to ensure that the comparisons are fair and all the benchmarks are implemented in an optimal way. We hope that these aspects are considered as merits for you, as we constantly experienced that comparisons in this field can be rather poor after having reproduced the results of several papers. As a final remark on this point, we do not see why it is better to use the same architecture to [2] and we wait for your enlightening answer.
> >
> > [1] Optimizing Neural Networks with Kronecker-factored Approximate Curvature. 2015
> >
> > [2] A scalable Laplace approximation for neural networks. 2018
> >
> > As a concluding remark, we thank you again for a very detailed review and we sincerely hope you understand the necessity of low rank approximation with this new revision.

---

> > > ### Comment · AnonReviewer4 · 2019-11-15
> > > **Thanks for your detailed rebuttal**
> > >
> > > Thanks for your detailed rebuttal and revisions, the paper looks much better now. And you rebuttal has resolved most of previous concerns.
> > >
> > > # Low-rank approximation
> > > You are right, low-rank approximation is necessary for sampling from the diag-corrected information form. But I feel it debatable that if you gain more from $D$ or you loss more from the sparsification.
> > >
> > > # $D$ being positive
> > > You mentioned that "However, with low-rank approximation and adding the diagonal correction afterwards, you can ensure that D remains strictly positive." Sorry I don't get it, why D is strictly positive? It is still possible that some of you low-rank diagonal terms are bigger than the true diagonal, isn't it?

---

> > > > ### Author Response · Authors · 2019-11-15
> > > > **Thank you for engaging in a discussion with us.**
> > > >
> > > >
> > > > Thank you so much for your further comments. We address your questions below.
> > > >
> > > > On Low-rank approximation: "if you gain more from $D$ or you loss more from the sparsification"
> > > >
> > > > This is a brilliant question. In short, you are correct - it is a hypothesis: keeping the diagonals exact while sparsifying the off-diagonals should result in a better estimates of model uncertainty (equivalently keeping the information content of a node exact while sparsifying the weak links between the nodes from a graphical interpretation of information matrix). Consequently, we have updated our experiments section where we discuss this point.
> > > >
> > > > On the other hand, we also point out that this is a well motivated hypothesis by practice. In SLAM literature (as a Bayesian tracking problem), so-called sparse information filters works under this hypothesis (map posterior in [1] is equivalent to the parameter posterior given the data; the recursion of Bayesian tracking can be generalized with a batch of data as well) and has been a compelling alternative to Kalman filters when the canonical form of full Gaussian distribution was intractable or inefficient. [1-5] are practical examples where this hypothesis worked well in practical applications.
> > > >
> > > > Now, in case of deep neural networks, our experiments verify this hypothesis. We find this valid as the spectrum (or eigenvalues) of information matrix tends to be sparse for DNNs (this is a major difference to SLAM problems and a reason why we introduced spectrum sparsification instead) and we find a sparse information form to perform well when compared to fully factorized and matrix normal distributions (as we use Laplace approximation, the approximate inference computations and the parameter posterior of a model are exactly the same; so only a main difference was the representation of model uncertainty).
> > > >
> > > > Despite these, we acknowledge that this is a limitation of our work. When the spectrum of information matrix is non-sparse, it is clearly debatable if this sparse information form can be an alternative to fully factorized and matrix normal distributions. This point has a strong connection to information geometry and Bayesian deep learning, and we feel there is a lack of foundations from a theoretic perspective. Nevertheless, we hope that our work is a stepping stone towards the goal of providing useful sparse expression to represent model uncertainty of deep neural networks.
> > > >
> > > > On $D$ being positive.
> > > >
> > > > Our apologies if this point was not clear. Let us rephrase: with a low-rank approximation and adding the diagonal correction afterwards, you can ensure that D remains strictly positive by choosing rank K (resulting in rank L=J+K) so that $[(U_{A_{1:a}} \otimes U_{G_{1:g}})\Lambda_{1:L} (U_{A_{1:a}} \otimes U_{G_{1:g}})]_{ii}^T < \mathbb{E} \left [ \delta \theta_i^2 \right ]$ in accordance of Lemma 3. This introduces another hyperparameter but it can be made automatic since the low rank approximation and diagonal correction are applied in off-line without involving data. We have also discuss this in a paragraph below Lemma 3.
> > > >
> > > > We thank you again for engaging in a discussion with us. Please let us know if you have more valuable feedback. We will incorporate them.
> > > >
> > > > References:
> > > >
> > > > [1] Simultaneous Localization and Mapping With Sparse Extended Information Filters.  2004.
> > > >
> > > > [2] Multi-Robot SLAM With Sparse Extended Information Filters. 2003.
> > > >
> > > > [3] Square root SAM: Simultaneous location and mapping via square root information smoothing. 2006.
> > > >
> > > > [4] Exactly sparse delayed-state filters. 2005
> > > >
> > > > [5] Simultaneous Localization and Mapping (SLAM): Part II. 2006 (for a survey).

---

### Official Review · AnonReviewer3 · 2019-11-02
**Official Blind Review #3**

**Rating:** 3

**Review:**


The paper studies the Laplace approximation for Bayesian inference of a neural network. Specifically, it proposes a diagonal correction and a further low-rank approximation to the Kronecker-factored eigenbasis for more accurate approximation of the Fisher information matrix and better scalablility, respectively. The proposed diagonal correction is shown to have a smaller residual error in F-norm. Experiments are given to show that the proposed Laplace approximation makes more accurate uncertainty estimations.

The paper makes a certain contribution to existing Laplace approximations for the task in terms of accuracy and scalability. However, it is incremental and the novelty is a bit low, compared to many recent closely related works, for example,

Optimizing Neural Networks with Kronecker-factored Approximate Curvature. 2015
Practical Gauss-Newton Optimisation for Deep Learning. 2017
Fast Approximate Natural Gradient Descent in a Kronecker-factored Eigenbasis. 2018
A scalable Laplace approximation for neural networks. 2018
Eigenvalue Corrected Noisy Natural Gradient. 2018


**Experience Assessment:**

I have published one or two papers in this area.

**Review Assessment: Checking Correctness Of Derivations And Theory:**

I assessed the sensibility of the derivations and theory.

**Review Assessment: Checking Correctness Of Experiments:**

I assessed the sensibility of the experiments.

**Review Assessment: Thoroughness In Paper Reading:**

I read the paper at least twice and used my best judgement in assessing the paper.

---

> ### Author Response · Authors · 2019-11-11
> **We hope for further discussions with you.**
>
> Thank you for the time you spent reviewing our paper and we sincerely hope for further discussions with you. We address your comments below.
>
> You comment: "The paper makes a certain contribution to existing Laplace approximations for the task in terms of accuracy and scalability. However, it is incremental and the novelty is a bit low, compared to many recent closely related works, for example, [1-5]."
>
> - Our proposed contributions have not been introduced in references you mention ([1-5]) as a fact, and these are modest increment similar to all research papers (we argue it is a matter of presentation).
>
> 1. Our contributions namely (a) diagonal correction to eigenbasis, (b) a low rank approximation preserving Kronecker structure in eigenvectors, (c) an algorithm to achieve the previous point, and (d) derivation of analytical sampler, have not appeared in the references you mention ([1-5]). We get your feelings that point (a) heavily builds on existing works but other points (b-d) are not. Furthermore, all these points are sensible and non-obvious, and empirical results confirm that the utility of our contributions are significant in terms of "accuracy and scalabilty" as you acknowledge. Lastly, all these contributions lead to representing model uncertainty in sparse information form, which conceptually, is different to references [1-5] and literature of Bayesian Deep learning .
>
> 2. We understand your sentiment: we start by adding a diagonal correction term to [3] for the framework of [4]. However, in a high level abstraction, all the references you mention can be phrased in a way that appear incremental, despite being highly influential works. Examples: (a) [1] extends [6] by introducing Kronecker product for the Fisher estimates. (b) [2] extends [1] to Gauss Newton instead of natural gradient. (c) [3] extends [1] by introducing a re-scaling term in eigenbasis. (d) [4] extends [7] by employing [2] for Laplace approximation. (e) [5] extends [8] by employing [3] for variational inference. In short, it is a matter of presentation (which we prioritized readers understandings rather than sounding completely new).
>
> [1] Optimizing Neural Networks with Kronecker-factored Approximate Curvature. 2015
>
> [2] Practical Gauss-Newton Optimisation for Deep Learning. 2017
>
> [3] Fast Approximate Natural Gradient Descent in a Kronecker-factored Eigenbasis. 2018
>
> [4] A scalable Laplace approximation for neural networks. 2018
>
> [5] Eigenvalue Corrected Noisy Natural Gradient. 2018
>
> [6] Natural Gradient Works Efficiently in Learning. 1998
>
> [7] A Practical Bayesian Framework for Backpropagation Networks. 1992
>
> [8] Noisy Natural Gradient as Variational Inference. 2018
>
> As a concluding remark, we sincerely hope to hear more from you in what you meant by "compared to many recent closely related works".

---

### Official Review · AnonReviewer5 · 2019-11-04
**Official Blind Review #5**

**Rating:** 6

**Review:**

The submitted paper presents a method of approximating the posterior distribution over the DNN parameters based on a Laplace Approximation scheme. It extends on the previous work by adding a diagonal correction term to the Kronecker-factored eigenbasis and also suggests a low-rank representation of Kronecker-factored eigendecomposition. Empirical evaluations were done to show that the proposed method has a more accurate uncertainty estimation compared to the previous work.

Overall, the paper is well-organized and easy to follow, although mathematical notations are not consistent throughout the paper. It is well-referenced, and the derivations look mostly correct (explained in minor comments). The main idea of the paper is convincing and well-motivated. To my knowledge, the proposed method of adding a correction term has not been introduced before. However, it is more of an incremental contribution to the existing works. In that sense, I am slightly concerned that its novelty is limited.

The experiments are not comprehensive. For the toy regression problem, a comparison to Hamiltonian Monte Carlo would be more informative. Moreover, it would be helpful to report the comparison with factorized variational methods (e.g. Graves, 2011) and experiment on modern architectures. I would be also interested to see what the additional time complexity by adding a diagonal correction term is and how more efficient it gets by low-rank approximation in experimental details.

Minor comments:
*Page 4: It would have been easier to understand if there had been a notational distinction between exact eigenbasis and K-FAC eigenbasis in defining V. Also, “In equation equation 10” -> “In equation 10”.
*Page 7: In Lemma 2, the order in describing the low-rank estimate does not appear to be correct. Also, in Lemma 4, shouldn’t there be a hat on I_{efb} and I_{kfac}?
*Page 9: The colour scheme in Figure 3 looks visually harder to read.
*Page 14: In equation 15 and 16, there should be a bracket for (2 \pi), and in Appendix B, some variables (e.g. p, k, R) are left unexplained.
*Page 14, 15: In equation 23 and proposition 1, \mathcal is missing, and vec operator is missing in equation 22.
*Page 3: In equation 2 and 4, is it A_{i-1} or A_i?
*Page 16: In equation 27, the dimension of X \in \mathcal{R}^{m \times m} seems incorrect. Also, in the last line, doesn’t X \odot D have different dimensions (similarly in equation 22)?

**Experience Assessment:**

I have published one or two papers in this area.

**Review Assessment: Checking Correctness Of Derivations And Theory:**

I carefully checked the derivations and theory.

**Review Assessment: Checking Correctness Of Experiments:**

I assessed the sensibility of the experiments.

**Review Assessment: Thoroughness In Paper Reading:**

I read the paper thoroughly.

---

> ### Author Response · Authors · 2019-11-11
> **Thank you for very useful comments**
>
> Thank you for very useful comments, and your review has improved our paper significantly. We start by addressing your main points.
>
> You comment: "Overall, the paper is well-organized and easy to follow, although mathematical notations are not consistent throughout the paper."
>
> - We sincerely apologize for our careless mistakes and we have made sure that the notations are consistent in the new revision. Furthermore, we have included several measures such as adding figures with notations so that these aspects are not confusing for the readers.
>
> You comment: "However, it is more of an incremental contribution to the existing works. In that sense, I am slightly concerned that its novelty is limited".
>
> - "Adding a correction term" is only a small part of the paper:
>
> We agree with your point that "adding a correction term" is an incremental contribution and novelty is limited. However, "adding a correction term" is only a small part of the paper. In fact, "proposing a low-rank Kronecker factored eigenbasis where we preserve Kronecker structure in eigenvectors", "devising an algorithm to achieve this", and "derivation of an analytical sampler" are the main technical/algorithmic contributions of the paper, that have not been introduced before. As a result, we show a novel representation of model uncertainty (sparse information formulation with low-rank Kronecker eigenbasis plus diagonal structure) without having to resort to fully factorized or matrix normal distributions. The utility of our approach is also demonstrated as we "show that the proposed method has a more accurate uncertainty estimation". We have improved our presentation style in the new revision.
>
> You comment: "For the toy regression problem, a comparison to Hamiltonian Monte Carlo would be more informative. Moreover, it would be helpful to report the comparison with factorized variational methods (e.g. Graves, 2011) and experiment on modern architectures."
>
> - This is a good idea clearly and we promise to add this before the revision period ends. On the modern architectures though, our experimental design has been chosen to show and validate our approach. In particular, the choice of architecture ensures that low-rank approximation is necessary (now illustrated in figure 1 of the revision).
>
> You comment: "I would be also interested to see what the additional time complexity by adding a diagonal correction term is and how more efficient it gets by low-rank approximation in experimental details."
>
> - This is a brilliant idea and we promise to add an empirical results in the current revision. We comment on time complexity of adding a diagonal correction term for the inference part. As a short note, all the computations required for adding a diagonal correction term does not involve data and are thus, off-line. Please find an overview in the last paragraph of section 2.3.
>
> On your minor comments:
>
> "Page 4: It would have been easier to understand if there had been a notational distinction between exact eigenbasis and KFAC eigenbasis in defining V. Also, “In equation equation 10” $->$ “In equation 10”."
>
> - This suggestion has been incorporated. We apologize for our careless mistakes.
>
> "Page 7: In Lemma 2, the order in describing the low-rank estimate does not appear to be correct. Also, in Lemma 4, should not there be a hat on $I_{efb}$ and $I_{kfac}$?"
>
> - We apologize for the confusion (indeed there should be a hat). We have made this more clear in the new revision.
>
> "Page 9: The colour scheme in Figure 3 looks visually harder to read."
>
> - We promise to change this either before the end of revision period.
>
> "Page 14: In equation 15 and 16, there should be a bracket for (2 $\pi$), and in Appendix B, some variables (e.g. p, k, R) are left unexplained."
>
> - We apologize for the confusion and we have made this clear in the new revision.
>
> "Page 14, 15:".
>
> - We apologize for the confusion and we have made this clear in the new revision.
>
> "Page 3: In equation 2 and 4, is it $A_{i-1}$ or $A_i$?"
>
> - Yes you are correct. We apologize the the carelessness and this is made clear in the new revision.
>
> "Page 16: In equation 27, the dimension of X $\in \mathcal{R}^{m \times m}$ seems incorrect. Also, in the last line, doesn’t $X \odot D$ have different dimensions (similarly in equation 22)?"
>
> - We apologize for the confusion and we have made this clear in the new revision. In fact, we have presented this section with a better presentation style.
>
> We thank you again for a very detailed review. Your comments have significantly improved our paper.

---

> > ### Author Response · Authors · 2019-11-15
> > **Please find the experiments you have requested in the new revision.**
> >
> > Please find the experiments you have requested in the new revision. In particular:
> >
> > 1. We have added HMC in our toy experiments and revised the description of our experiment section. We have also added Bayes by Backprop (instead of Graves 2011) for this experiment which is also based on variational inference and fully factorized Gaussian. A main reason was that it was slightly more recent than Graves 2011 and often compared baselines (which we hope you agree).
> >
> > 2. We have added a part that shows reduction in complexity. Please find the related discussions in section 4.2 of new revision.
> >
> > Your review has improved our paper significantly! Thank you again and please let us know if you have another valuable feedback.

---

### Author Response · Authors · 2019-11-15
**A general response to all the reviewers.**



We sincerely thank all the reviewers for their time and efforts. The paper has been thoroughly revised in light of your thoughtful feedback. In this post, we attempt to shortly summarize the main points of paper and changes in the new revision.

A summary:

This work presents a sparse information form of multivariate normal distribution (MND) to represent model uncertainty of neural networks, in oppose to fully factorized Gaussian and matrix normal distributions. With Laplace Approximation as a backbone, we point out that model uncertainty can be inferred in information form of MND by adding a diagonal correction term. Despite being more general formulation, MND suffers from its inherent intractable complexity. Consequently, we demonstrate a solution by newly introducing (a) a sparse information form, (b) a sparsification algorithm, and (b) its tightly coupled analytical sampler. Lastly, we theoretically and empirically show state-of-the-art performance.

As a result, we demonstrate a way to tackle non-trivial challenges that are associated with intractability of MND which presents itself into a novel expression for the model uncertainty. We firmly believe that Bayesian Deep Learning need better representation of the parameter posterior along with better approximate Bayesian inference. In this sense, our work can be a stepping stone towards this direction.



The current revision contains following key changes:

(a) notations are made consistent [R4 and R5].

(b) a cleaner presentation style including figures and examples that better explains the concept [R4].

(c) a clearer statement on the main novelty in related works [R3 and R5].

(d) moved our theoretical results to appendix [R2].

(e) expanded the summary of main contributions, pointing at specific parts [R2, R4].

(g) additional experiments and related discussions to address reviewer's feedback [R4 and R5].

In particular, Hamiltonian Monte Carlo (Neal 1996) and Bayes by Backpropagation (Blundell 2015) have been added as benchmarks to include a sampling method (as a better ground truth) and a fully factorized variational inference method. Moreover, we have added more ablation studies. Lastly, we also show the reduction of complexity due to the low rank approximation for our classification experiments.

We individually address the reviewers in more detail below.

---

### Decision · Program_Chairs · 2019-12-19

**Decision:**

Reject

**Comment:**

This paper presents a variant of recently developed Kronecker-factored approximations to BNN posteriors. It corrects the diagonal entries of the approximate Hessian, and in order to make this scalable, approximates the Kronecker factors as low-rank.

The approach seems reasonable, and is a natural thing to try. The novelty is fairly limited, however, and the calculations are mostly routine. In terms of the experiments: it seems like it improved the Frobenius norm of the error, though it's not clear to me that this would be a good measure of practical effectiveness. On the toy regression experiment, it's hard for me to tell the difference from the other variational methods. It looks like it helped a bit in the quantitative comparisons, though the improvement over K-FAC doesn't seem significant enough to justify acceptance purely based on the results.

Reviewers felt like there was a potentially useful idea here and didn't spot any serious red flags, but didn't feel like the novelty or the experimental results were enough to justify acceptance. I tend to agree with this assessment.